# Atoh1 is required for the formation of lateral line electroreceptors and hair cells, whereas FoxG1 represses an electrosensory fate

Martin Minařík[1], Alexander S Campbell[1], Roman Franěk[2], Michaela Vazačová[2], Miloš Havelka[2], David Gela[2], Martin Pšenička[2], Clare VH Baker[1]*

[1]Department of Physiology, Development & Neuroscience, University of Cambridge, Cambridge, United Kingdom; [2]Faculty of Fisheries and Protection of Waters, Research Institute of Fish Culture and Hydrobiology, University of South Bohemia in České Budějovice, Vodňany, Czech Republic

**Abstract** In electroreceptive jawed fishes and amphibians, individual lateral line placodes form lines of neuromasts on the head containing mechanosensory hair cells, flanked by fields of ampullary organs containing electroreceptors—modified hair cells that respond to weak electric fields. Extensively shared gene expression between neuromasts and ampullary organs suggests that conserved molecular mechanisms are involved in their development, but a few transcription factor genes are restricted either to the developing electrosensory or mechanosensory lateral line. Here, we used CRISPR/Cas9-mediated mutagenesis in G0-injected sterlet embryos (*Acipenser ruthenus*, a sturgeon) to test the function of three such genes. We found that the 'hair cell' transcription factor gene *Atoh1* is required for both hair cell and electroreceptor differentiation in sterlet, and for *Pou4f3* and *Gfi1* expression in both neuromasts and ampullary organs. These data support the conservation of developmental mechanisms between hair cells and electroreceptors. Targeting ampullary organ-restricted *Neurod4* did not yield any phenotype, potentially owing to redundancy with other *Neurod* genes that we found to be expressed in sterlet ampullary organs. After targeting mechanosensory-restricted *Foxg1*, ampullary organs formed within neuromast lines, suggesting that FoxG1 normally represses their development, whether directly or indirectly. We speculate that electrosensory organs may be the 'default' developmental fate of lateral line primordia in electroreceptive vertebrates.

*For correspondence:
cvhb1@cam.ac.uk

Competing interest: The authors declare that no competing interests exist.

## Editor's evaluation

Many aquatic vertebrates have a lateral line system comprising both mechanosensory and electrosensory receptor cells. Using a gene knockout approach in the sterlet, this study convincingly demonstrates that the development of both types of receptor cells depends on the transcription factor Atoh1, it further shows that another transcription factor, FoxG1, promotes mechanoreceptor but represses electroreceptor development. Particularly interesting from the evolutionary perspective is the resulting hypothesis of the authors that electrosensory organs may be the 'default' developmental fate in electroreceptive vertebrates. Overall, the study provides fundamental new insights into sensory development and evolution.

## Introduction

Mechanosensory hair cells in different compartments of the inner ear transduce fluid movements for hearing and balance (*Moser et al., 2020*; *Caprara and Peng, 2022*; *Mukhopadhyay and Pangrsic, 2022*). In fishes and aquatic-stage amphibians, hair cells are also found in lateral line neuromasts in the skin, which are stimulated by local water movement (*Mogdans, 2019*; *Webb, 2021*). Supporting cells in neuromasts differentiate into hair cells for homeostasis and repair (*Kniss et al., 2016*; *Lush et al., 2019*) and neuromasts are easily accessible in lines over the head and trunk, making the zebrafish lateral line an excellent model for hair cell development and regeneration (*Kniss et al., 2016*; *Nicolson, 2017*; *Pickett and Raible, 2019*).

The lateral line system of zebrafish (a cyprinid teleost) is purely mechanosensory, as is the lateral line system of the other main anamniote lab model, the frog *Xenopus*. However, in many other vertebrates, the lateral line system also has an electrosensory division (*Bullock et al., 1983*; *Baker et al., 2013*; *Crampton, 2019*). In electroreceptive non-teleost jawed vertebrates, some or all of the neuromast lines on the head are flanked by fields of ampullary organs containing electroreceptors, which respond to weak cathodal stimuli such as the electric fields surrounding other animals in water (*Bodznick and Montgomery, 2005*; *Crampton, 2019*; *Leitch and Julius, 2019*; *Chagnaud et al., 2021*). Electroreception is mediated by voltage-gated calcium channels in the apical membrane (*Bodznick and Montgomery, 2005*; *Leitch and Julius, 2019*). The voltage sensor was recently identified in cartilaginous fishes as the L-type voltage-gated calcium channel Ca$_v$1.3 (*Bellono et al., 2017*; *Bellono et al., 2018*), whose pore-forming alpha subunit is encoded by *Cacna1d*. Ca$_v$1.3 is also required for synaptic transmission at hair-cell ribbon synapses (*Moser et al., 2020*; *Mukhopadhyay and Pangrsic, 2022*).

Electroreceptors, like hair cells, have an apical primary cilium and basolateral ribbon synapses with lateral line afferents (*Jørgensen, 2005*; *Baker, 2019*). However, in contrast to the highly ordered, stepped array of apical microvilli ('stereocilia') that forms the 'hair bundle' critical for hair-cell mechanotransduction (*Caprara and Peng, 2022*), electroreceptors in many species (for example, cartilaginous fishes; ray-finned paddlefishes and sturgeons) lack apical microvilli altogether (*Jørgensen, 2005*; *Baker, 2019*). Electroreceptors in other species have a few apical microvilli, while the electroreceptors of the amphibian axolotl have around 200 microvilli surrounding an eccentrically positioned primary cilium (*Jørgensen, 2005*; *Baker, 2019*). Indeed, axolotl electroreceptors were described as "remarkably similar to immature hair cells" (*Northcutt et al., 1994*). Thus, despite their shared function, the apical surface of electroreceptors (where voltage-sensing occurs; *Bodznick and Montgomery, 2005*; *Leitch and Julius, 2019*) varies considerably across different vertebrate groups.

Fate-mapping experiments have shown that neuromasts, ampullary organs (where present) and their afferent neurons all develop from a series of pre-otic and post-otic lateral line placodes on the embryonic head (*Northcutt, 1997*; *Piotrowski and Baker, 2014*; *Baker, 2019*). In electroreceptive jawed vertebrates, lateral line placodes elongate to form sensory ridges that eventually fragment: neuromasts differentiate first, in a line along the centre of each ridge, and ampullary organs (if present) form later, in fields on the flanks of the ridge (*Northcutt, 1997*; *Piotrowski and Baker, 2014*; *Baker, 2019*). The lateral line primordia of electroreceptive vertebrates therefore provide a fascinating model for studying the formation of different sensory cell types and organs. What molecular mechanisms control the formation within the same primordium of mechanosensory neuromasts containing hair cells, versus electrosensory ampullary organs containing electroreceptors?

To gain molecular insight into electroreceptor development, we originally took a candidate-gene approach, based on genes known to be important for neuromast and/or hair cell development. This enabled us to identify a variety of genes expressed in developing ampullary organs as well as neuromasts, in embryos from the three major jawed-vertebrate groups, that is, cartilaginous fishes (lesser-spotted catshark, *Scyliorhinus canicula*, and little skate, *Leucoraja erinacea*; *O'Neill et al., 2007*; *Gillis et al., 2012*); lobe-finned bony fishes/tetrapods (a urodele amphibian, the axolotl, *Ambystoma mexicanum*; *Modrell and Baker, 2012*); and ray-finned bony fishes (a chondrostean, the Mississippi paddlefish, *Polyodon spathula*; *Modrell et al., 2011a*; *Modrell et al., 2011b*; *Modrell et al., 2017b*). We also took an unbiased discovery approach using differential bulk RNA-seq in late-larval paddlefish, which yielded a dataset of almost 500 genes that were putatively enriched in lateral line organs (*Modrell et al., 2017a*). Validation by in situ hybridisation (ISH) of a subset of candidates from this dataset suggested that conserved molecular mechanisms were involved in hair cell and electroreceptor

development, and that hair cells and electroreceptors were closely related physiologically (*Modrell et al., 2017a*). For example, developing ampullary organs express the key 'hair cell' transcription factor genes *Atoh1* and *Pou4f3* (see *Roccio et al., 2020*; *Iyer and Groves, 2021*), and genes essential for the function of hair cell ribbon synapses, including the voltage-gated calcium channel gene *Cacna1d*, encoding Ca$_v$1.3 (*Modrell et al., 2017a*). We also identified a handful of genes expressed in developing ampullary organs but not neuromasts, including two electroreceptor-specific voltage-gated potassium channel subunit genes (*Kcna5* and *Kcnab3*) and a single transcription factor gene, *Neurod4* (*Modrell et al., 2017a*).

Up to that point, we had reported the shared expression of fifteen transcription factor genes in both ampullary organs and neuromasts, but only one transcription factor gene with restricted expression, namely, electrosensory-restricted *Neurod4* (*Modrell et al., 2011a*; *Modrell et al., 2011b*; *Modrell et al., 2017a*). More recently (*Minařík et al., 2024*), we used the late-larval paddlefish lateral line organ-enriched dataset (*Modrell et al., 2017a*), as well as a candidate gene approach, to identify 23 more transcription factor genes expressed within developing lateral line organs in paddlefish and/or in a related, more experimentally tractable chondrostean, the sterlet (*Acipenser ruthenus*, a small sturgeon; for example, *Chen et al., 2018*; *Baloch et al., 2019*; *Stundl et al., 2022*). Twelve of these transcription factor genes—including *Gfi1*—were expressed in both ampullary organs and neuromasts (*Minařík et al., 2024*). Thus, developing ampullary organs, as well as neuromasts, express the three 'hair cell' transcription factor genes—*Atoh1*, *Pou4f3*, and *Gfi1*—whose co-expression is sufficient to drive postnatal mouse cochlear supporting cells to adopt a 'hair cell-like' fate, albeit not to form fully mature hair cells (*Roccio et al., 2020*; *Iyer and Groves, 2021*; *Chen et al., 2021*; *Iyer et al., 2022*). We also identified six novel ampullary organ-restricted transcription factor genes and the first-reported mechanosensory-restricted transcription factor genes (*Minařík et al., 2024*). One of the five mechanosensory-restricted transcription factor genes was *Foxg1*, which was expressed and maintained in the central region of sensory ridges where lines of neuromasts form, although excluded from hair cells (*Minařík et al., 2024*).

Here, we used CRISPR/Cas9-mediated mutagenesis in G0-injected sterlet embryos to investigate the function in lateral line organ development of *Atoh1*, electrosensory-restricted *Neurod4* and mechanosensory-restricted *Foxg1* (for reference, **Figure 1** shows the normal expression patterns of these genes). We report that *Atoh1* is required for the formation of electroreceptors, as well as hair cells. We did not see any phenotype after targeting ampullary organ-restricted *Neurod4*, potentially owing to redundancy with other *Neurod* family members that we found to be expressed in sterlet ampullary organs (and neuromasts). Targeting mechanosensory-restricted *Foxg1* resulted in a striking phenotype: the ectopic formation of ampullary organs within neuromast lines, and in some cases the fusion of ampullary organ fields that normally develop either side of a line of neuromasts. In addition, sections of neuromast lines were often missing mosaically, supporting a direct role for FoxG1 in neuromast development, as recently reported in zebrafish (*Bell et al., 2024*). However, the presence of ectopic ampullary organs within neuromast lines in *Foxg1* crispants suggests the unexpected but intriguing hypothesis that ampullary organs may be the 'default' developmental fate for lateral line sensory ridges, and that this is repressed by FoxG1, allowing neuromasts to form instead.

## Results
### CRISPR/Cas9-mediated mutagenesis in G0-injected sterlet embryos

To test gene function during lateral line organ development, we optimised CRISPR/Cas9-mediated mutagenesis in G0-injected sterlet embryos, building on established protocols for axolotl (*A. mexicanum*; *Flowers et al., 2014*; *Fei et al., 2018*), newt (*Pleurodeles waltl*; *Elewa et al., 2017*), and sea lamprey (*Petromyzon marinus*; *Square et al., 2015*; *York et al., 2019*; *Square et al., 2020*), whose eggs are all large (1–2 mm in diameter) and easy to microinject at the 1–2 cell stage. Ovulated sterlet eggs are very large (roughly 2.5 mm in diameter; *Lenhardt et al., 2005*) and undergo a holoblastic cleavage (*Dettlaff et al., 1993*). (Since this project started, CRISPR/Cas9-mediated mutagenesis in G0-injected sterlet embryos has been reported, including by two of us, RF and MP; *Chen et al., 2018*; *Baloch et al., 2019*; *Stundl et al., 2022*.) Analysis of microsatellite data had originally suggested that although a whole-genome duplication had occurred in the sterlet lineage, the sterlet was likely to be a functional diploid (*Ludwig et al., 2001*). Our sgRNAs were designed before the first chromosome-level

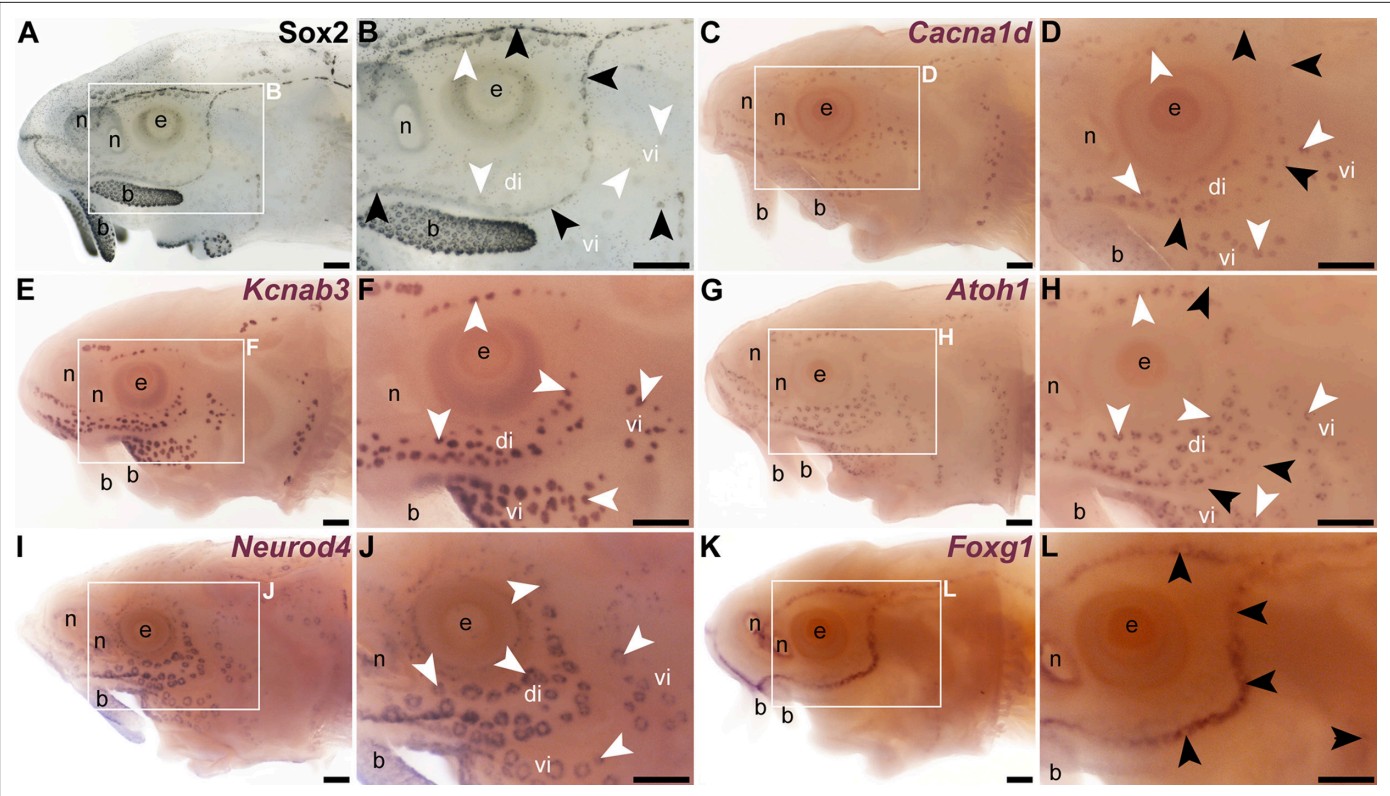

**Figure 1.** Wild-type gene expression patterns within late-larval sterlet lateral line organs. Detailed descriptions of the sterlet gene expression patterns shown for reference in this figure, including at earlier stages of lateral line development, have been published (*Minařík et al., 2024*), except for *Neurod4*. Sterlet are shown at late yolk-sac larval stages. Black arrowheads indicate examples of neuromasts; white arrowheads indicate examples of ampullary organs. (**A**, **B**) Immunostaining for the supporting cell marker Sox2 at stage 45 shows strong expression in neuromasts and much weaker expression in ampullary organs. Sox2 is also expressed in the nares, retina, and taste buds on the barbels and around the mouth. (**C**, **D**) In situ hybridisation (ISH) for the hair cell and electroreceptor marker *Cacna1d* at stage 45 shows expression in neuromasts and ampullary organs. Much weaker *Cacna1d* expression is seen in taste buds on the barbels; this is not always detectable. (**E**, **F**) ISH for the electroreceptor marker *Kcnab3* at stage 45 shows expression in ampullary organs only. (**G**, **H**) ISH for *Atoh1* at stage 42 shows expression in ampullary organs and, more weakly, in neuromasts. (**I**, **J**) ISH for *Neurod4* at stage 45 shows expression in ampullary organs but not neuromasts. *Neurod4* expression is also seen in taste buds on the barbels. (**K**, **L**) ISH for *Foxg1* at stage 45 shows expression is restricted to neuromast lines, though excluded from the centres of neuromasts where hair cells form (compare with Sox2 expression in supporting cells in panel **B**, and with *Cacna1d* expression in hair cells in panel **D**). *Foxg1* expression is also seen in the nares. Abbreviations: b, barbel; di, dorsal infraorbital ampullary organ field; e, eye; n, naris; vi, ventral infraorbital ampullary organ field. Scale bar: 200 µm.

The online version of this article includes the following figure supplement(s) for figure 1:

**Figure supplement 1.** Examples of successful disruption of sterlet *Neurod4* by CRISPR/Cas9-mediated mutagenesis in G0-injected embryos.

**Figure supplement 2.** Expression of *Neurod* family genes in late-larval sterlet lateral line organs.

sterlet genome was published (*Du et al., 2020*). Analysis of this genome, together with transcriptomic data, showed that approximately 70% of ohnologs (i.e., gene paralogs originating from the whole-genome duplication) had been retained and suggested a high level of functional tetraploidy (*Du et al., 2020*). We comment on this in relation to our experiments at the relevant points below.

We targeted the melanin-producing enzyme *tyrosinase* (*Tyr*) as a positive control, using eight different single-guide (sg) RNAs. *Table 1* shows the sgRNA target sequences (including two that were designed and recently published by *Stundl et al., 2022*) and the various combinations in which they were injected. *Figure 2A* shows the position of each sgRNA relative to the exon structure of the *tyrosinase* gene. Our sgRNAs were designed before chromosome-level sterlet genomes were available (*Du et al., 2020* and the 2022 reference genome: GCF_902713425.1). Searching the reference genome for *Tyr* showed that both *Tyr* ohnologs have been retained, on chromosomes 8 and 9, with 99.11% nucleotide identity in the coding sequence (98.81% amino acid identity). Our sgRNAs target both *Tyr* ohnologs equally.

**Table 1.** sgRNAs used in this study.

List of the genes targeted for CRISPR/Cas9-mediated mutagenesis, together with the target sequences and combinations of sgRNAs reported in this study. *Tyr* sgRNAs 7 and 8 (marked with an asterisk) were designed and published by *Stundl et al., 2022* as their *tyr* sgRNA 3 and *tyr* sgRNA 4, respectively.

| Target gene | sgRNA | Target sequence | PAM | Combinations used |
|---|---|---|---|---|
| *Tyr* | 1 | GGTGCCAAGGCAAAAACGCT | GGG | 1+2, 1+2+3+4 |
| | 2 | GATATCCCTCCATACATTAT | TGG | 1+2, 1+2+3+4 |
| | 3 | GATGTTTCTAAACATTGGGG | TGG | 1+2+3+4 |
| | 4 | GCTATGAATTTATTTTTTTC | AGG | 1+2+3+4 |
| | 5 | GCAAGGTATACGAAAGTTGA | CGG | 5+6 |
| | 6 | GATTGCAAGTTCGGCTTCTT | TGG | 5+6 |
| | 7* | GGTTAGAGACTTTATGTAAC | GGG | 7+8 |
| | 8* | GGCTCCATGTCTCAAGTCCA | AGG | 7+8 |
| *Atoh1* | 1 | GACCTTGTAAAAGATCGGAA | AGG | 1+2 |
| | 2 | GCTTGTCATTGTCAAATGAC | GGG | 1+2 |
| *Neurod4* | 1 | GGAGCGTTTCAAGGCCAGGC | GGG | 1, 1+2+3+4, 1+6 |
| | 2 | GTGAGCGTTCTCGCATGCAC | GGG | 2, 1+2+3+4 |
| | 3 | GCCTGGCCCACAACTACATC | TGG | 1+2+3+4, 3+4+5+6 |
| | 4 | GAGGGGCCCCGAGAAGCTGC | AGG | 1+2+3+4, 3+4+5+6 |
| | 5 | GTCTCCCCAGCCCTCCCTAC | GGG | 5, 3+4+5+6 |
| | 6 | GACAACCACTCCCCGGATTG | CGG | 1+6, 3+4+5+6 |
| | 7 | GACCCTGCGCAGGCTCTCCA | GGG | 7+9, 7+8+9 |
| | 8 | GCAGCTGGGTCCCCTGCTGA | CGG | 7+8+9 |
| | 9 | GGGGCCGTGTGCTCAGGGAT | GGG | 7+9, 7+8+9 |
| *Foxg1* | 1 | GAAACATCTTTTGCCCAACC | CGG | 1+2 |
| | 2 | TCTTCCGAGCAAGGTAACTC | GGG | 1+2 |
| | 3 | TGATGCTGAAGGACGACTTG | GGG | 3+4 |
| | 4 | CTGGCTCGTCCTCGGGCCGG | TGG | 3+4 |

Three different combinations of six of the eight *Tyr* sgRNAs, when injected at the 1–2-cell stage (2–4 sgRNAs pre-complexed with Cas9), each generated at least four larvae (hereafter 'crispants') with altered pigmentation phenotypes evaluated at stage 45 (*Dettlaff et al., 1993*), the onset of the transition to independent feeding. The other two *Tyr* sgRNAs failed to generate any phenotypes (*Supplementary file 1*). Excluding the *Tyr* sgRNAs that failed, at least some degree of pigment loss was seen in 63/111 *Tyr* crispants (56.8%) across nine independent batches. Examples of *Tyr* crispants with pigmentation phenotypes, plus a control, are shown in *Figure 2B–D*. The most efficient results were obtained by injecting 1-cell embryos with a preassembled mix of Cas9 protein plus two chemically modified sgRNAs (purchased from Synthego) against the target gene, and subsequently maintaining the embryos at room temperature for around 6 hours. The time from fertilisation to completion of the first cleavage is around 2–3 hours at room temperature, giving plenty of time for the Cas9/sgRNA complex to act before returning the embryos to colder temperatures for subsequent development. (Sterlet are cold-water fish and the optimum temperature for maintaining embryos for normal development is 16°C.)

Following embryo injection at the 1–2-cell stage with pre-complexed sgRNAs/Cas9 and fixation at stage 45, genomic DNA was extracted from the trunk/tail prior to analysis of the heads by ISH. The sgRNA-targeted region from trunk/tail genomic DNA was amplified by PCR for direct Sanger

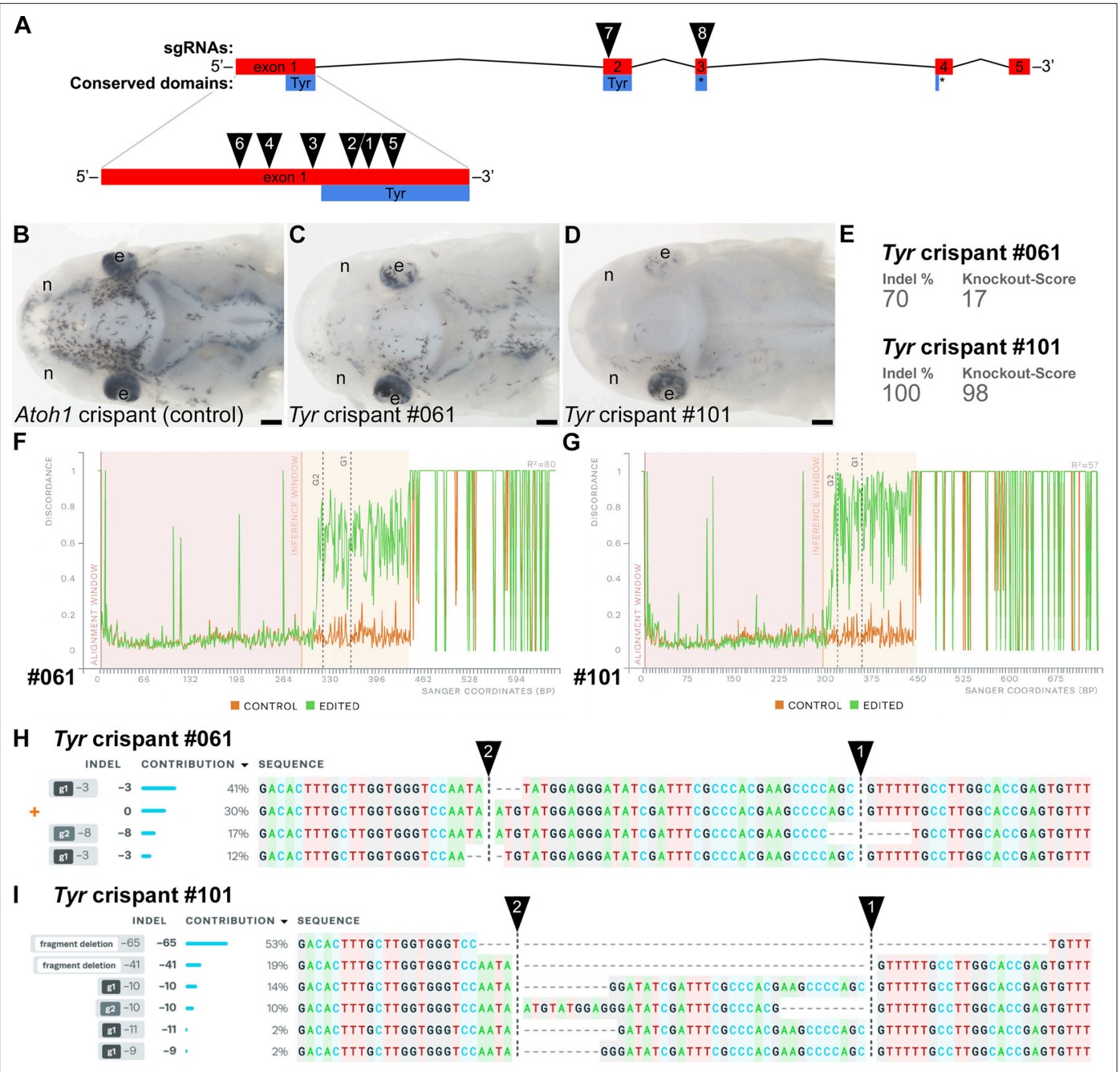

**Figure 2.** Examples of successful disruption of sterlet *tyrosinase* by CRISPR/Cas9-mediated mutagenesis in G0-injected embryos. (**A**) Schematic showing the exon structure (coding exons only) of the sterlet *tyrosinase* (*Tyr*) gene relative to conserved domains and the target sites of *Tyr* sgRNAs (*Table 1*). (**B**) Dorsal view of an *Atoh1* crispant at stage 45 as a control (*Atoh1* is not involved in melanin synthesis). Pigmented melanocytes are visible, particularly around the brain, and the eyes are fully pigmented. (**C, D**) Examples of *Tyr* crispants at stage 45 (both targeted with *Tyr* sgRNAs 1 and 2; see panel **A** and *Table 1*) with different degrees of pigment loss (compare with panel **B**). In both *Tyr* crispants, significantly fewer melanocytes are visible and the eyes show mosaic loss of pigment. The phenotype is stronger on the right side in both crispants, and stronger in crispant #101 (**D**) than in crispant #61 (**C**). (**E–I**) Outputs from Synthego's 'Inference of CRISPR Edits' (ICE) tool (*Conant et al., 2022*) applied to Sanger sequence data for the targeted region of genomic DNA extracted from the trunk/tail of each of the *Tyr* crispants shown in panel **C** and panel **D**. 'Indel %' (**E**) shows the percentage of insertions and/or deletions among the inferred sequences in the CRISPR-edited population. 'Knockout-Score' (**E**) indicates the proportion of indels that introduce a frameshift or are at least 21 bp long. Discordance plots (**F, G**) show the level of discordance between the edited sample Sanger trace file (green) and the control sample trace file (orange). Vertical dotted lines indicate the expected cut sites for the sgRNAs. The increase in discordance near the expected cut site indicates a successful CRISPR edit. Nucleotide sequences from the Sanger trace files and their inferred relative contributions to the edited mosaic population are shown in panel **H** and panel **I**. The expected cut sites are represented by vertical dotted lines. The wild-type sequence

*Figure 2 continued on next page*

*Figure 2 continued*

(0) is marked by an orange '+' symbol in panel **H**, but is absent in panel **I** because 100% of the sequence was edited in this crispant. Abbreviations: e, eye; n, naris. Scale bar: 200 μm.

sequencing and in silico analysis using Synthego's online 'Inference of CRISPR Edits' (ICE) tool (***Conant et al., 2022***; also see, for example, ***Uribe-Salazar et al., 2022***) to analyse the identity and frequency of edits of the target gene. Although our genotyping primers were designed before chromosome-level sterlet genomes were available, comparison with the reference genome showed no mismatches against either of the two ohnologs. Genotyping and ICE analysis (***Conant et al., 2022***) of tails from individual *Tyr* crispants showed successful disruption of the *Tyr* gene (***Figure 2E–I*** show examples of successful disruption of *Tyr*; the genotyping data were consistent with the primers amplifying both ohnologs).

We note that our genotyping results have shown that most crispants analysed, across all genes targeted, have shown some degree of targeted mutagenesis in the trunk/tail, with a range of deletion sizes. Although phenotypes from the initial spawning seasons were almost always highly mosaic, suggesting mutations occurred later in development, following optimisation a proportion of crispants showed complete unilateral and occasionally bilateral phenotypes. Such phenotypes suggest that mutation occurred in one cell at the 2-cell stage (unilateral phenotype) or even as early as the 1-cell stage (bilateral phenotype). Some degree of mosaicism can be useful, however, as the unaffected tissue provides an internal control for the normal expression of the gene being examined by ISH.

## Targeting *Atoh1* resulted in the mosaic absence of hair cells and electroreceptors

We targeted *Atoh1* for CRISPR/Cas9-mediated mutagenesis by injecting sterlet embryos at the 1–2-cell stage with Cas9 protein pre-complexed with two sgRNAs targeting *Atoh1* (***Table 1***, ***Figure 3—figure supplement 1A***). Our sgRNAs were designed before chromosome-level sterlet genomes were available (***Du et al., 2020*** and the 2022 reference genome: GCF_902713425.1). Searching the reference genome for *Atoh1* showed that both *Atoh1* ohnologs have been retained, on chromosomes 1 and 2, with 91.41% nucleotide identity (and 84.02% amino acid identity) in the coding sequence. The copy on chromosome 2 encodes a shorter version of the protein with a four amino acid deletion near the N-terminus (E14_G17del). Our *Atoh1* sgRNA 1 (***Table 1***, ***Figure 3—figure supplement 1A***) has a one-base mismatch to the shorter *Atoh1* gene on chromosome 2, in position 3 of the target sequence (PAM-distal), which is unlikely to prevent successful targeting (***Wu et al., 2014***). Our *Atoh1* sgRNA 2 (***Table 1***, ***Figure 3—figure supplement 1A***) has a two-base mismatch to the longer *Atoh1* gene on chromosome 1, in positions 1 and 2 of the target sequence (PAM-distal), which is also unlikely to prevent successful targeting. Thus, we expect our sgRNAs to target both *Atoh1* ohnologs.

The *Atoh1* crispants were raised to stage 45 (the onset of independent feeding, around 14 days post-fertilisation), together with *Tyr*-targeted siblings/half-siblings as controls (eggs were fertilised in vitro with a mix of sperm from three different males). Genotyping and ICE analysis (***Conant et al., 2022***) were performed on tails from individual *Atoh1* crispants targeted with this pair of sgRNAs (***Figure 3—figure supplement 1*** shows examples). Our genotyping primers were designed before chromosome-level sterlet genomes were available; comparison with the reference genome showed two mismatches in the forward primer against the ohnolog on chromosome 2, and the genotyping data were consistent with the primers amplifying the chromosome-1 ohnolog only. Thus, we could not determine whether the chromosome 2 ohnolog was disrupted. However, the genotyping data showed successful disruption of the *Atoh1* gene on chromosome 1 (examples are shown in ***Figure 3—figure supplement 1F–I***).

In *Tyr* control crispants, ISH for the hair cell and electroreceptor marker *Cacna1d* (***Modrell et al., 2017a***; also see ***Minařík et al., 2024***), which is a direct Atoh1 target gene in mouse cochlear hair cells (***Jen et al., 2022***), revealed no obvious lateral line organ phenotype (***Figure 3A–D***; n=0/24 across five independent batches; ***Supplementary file 1***). Even in wildtype larvae, the number of ampullary organs in individual fields varies considerably at stage 45, so ampullary organ number was not in

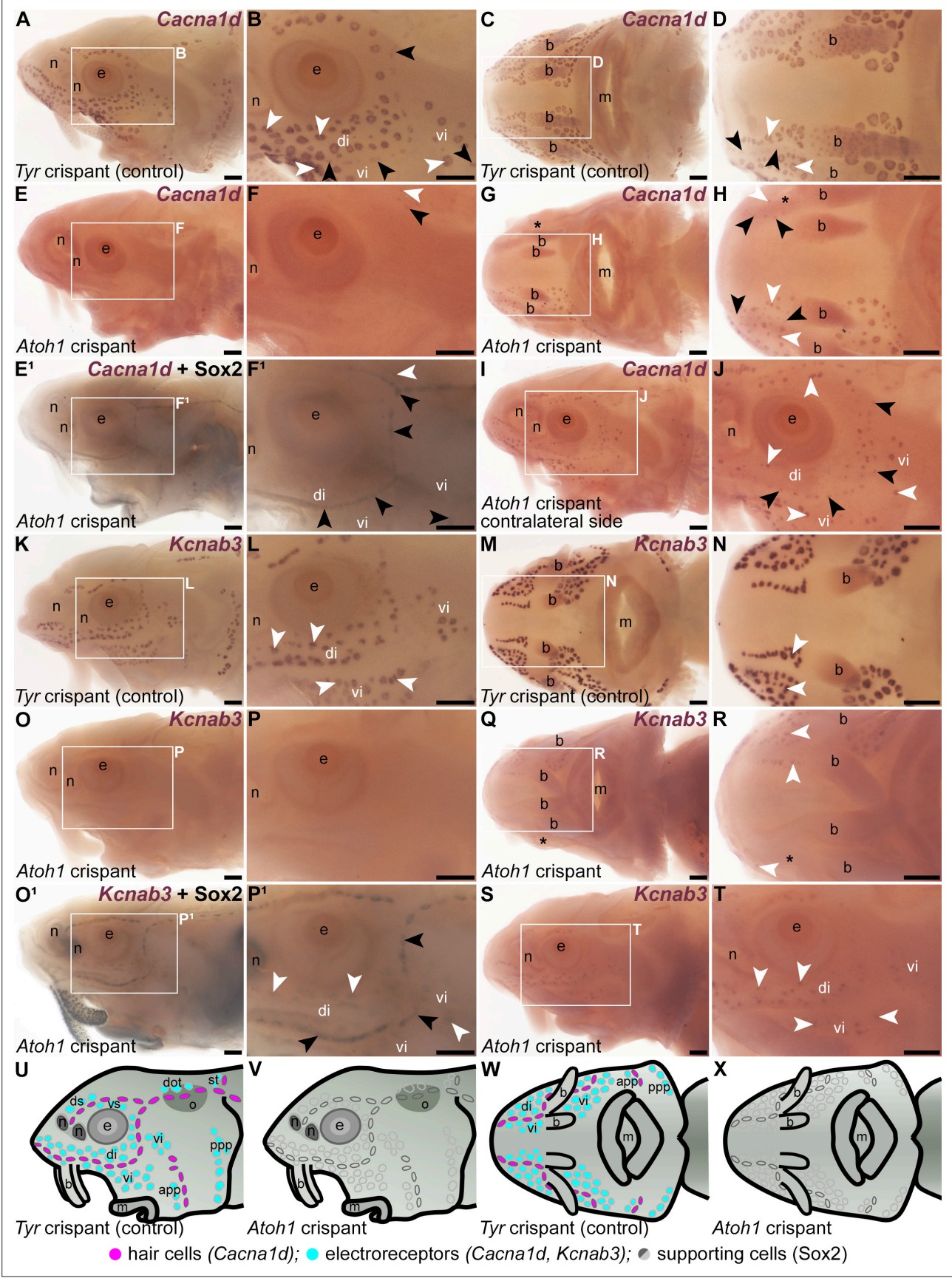

*Figure 3 continued*

**Figure 3.** *Atoh1* is required for the differentiation of lateral line hair cells and electroreceptors. Sterlet crispants at stage 45 after in situ hybridisation (ISH) for the hair cell and electroreceptor marker *Cacna1d*, or the electroreceptor-specific marker *Kcnab3*. Black arrowheads indicate examples of neuromasts; white arrowheads indicate examples of ampullary organs. (**A–D**) In a control *Tyr* crispant, *Cacna1d* expression shows the normal distribution of hair cells in lines of neuromasts, and electroreceptors in fields of ampullary organs flanking the neuromast lines (lateral view: **A, B**; ventral view: **C, D**). (**E–J**) In an *Atoh1* crispant (from a different batch to the *Tyr* crispant shown in **A–D**), *Cacna1d* expression is absent on the left side of the head (**E, F**), except for a few isolated organs in the otic region and on the operculum. Post-ISH Sox2 immunostaining (**E¹, F¹**) shows that neuromast supporting cells are still present; however, the signal is too weak to show ampullary organs. A ventral view (**G, H**) and a lateral view of the right side of the head (**I, J**; image flipped horizontally for ease of comparison) reveal a unilateral phenotype, with *Cacna1d*-expressing hair cells and electroreceptors mostly absent from the left side only of the ventral rostrum (asterisk in **G, H**) and present on the right side of the head (**I, J**). (**K, L**) Lateral view of a control *Tyr* crispant after ISH for *Kcnab3*, showing the position of electroreceptors in ampullary organs. (**M, N**) Ventral view of another *Tyr* crispant showing *Kcnab3* expression in ampullary organs. (**O–P¹**) Lateral view of an *Atoh1* crispant in which *Kcnab3* expression is absent from ampullary organs. Post-ISH Sox2 immunostaining (**O¹, P¹**) shows that supporting cells are still present in neuromasts (strong staining) and can also be detected in ampullary organs (much weaker staining). (**Q–T**) A different *Atoh1* crispant after ISH for *Kcnab3*, shown in ventral view (**Q, R**: compare with **M, N**) and lateral view (**S, T**). *Kcnab3* expression reveals a unilateral phenotype: *Kcnab3*-expressing electroreceptors are mostly absent from the right side (asterisk) but present on the left side. (**U–X**) Schematic representation of cranial lateral line organs in a stage 45 control *Tyr* crispant (lateral view, **U**; ventral view, **W**) versus a severe *Atoh1* crispant in which supporting cells (grey outlines) are present but all hair cells and electroreceptors are missing (lateral view, **V**; ventral view, **X**). Abbreviations: app, anterior preopercular ampullary organ field; b, barbel; di, dorsal infraorbital ampullary organ field; dot, dorsal otic ampullary organ field; ds, dorsal supraorbital ampullary organ field; e, eye; m, mouth; n, naris; o, otic capsule; ppp, posterior preopercular ampullary organ field; st, supratemporal ampullary organ field; vi, ventral infraorbital ampullary organ field; vs, ventral supraorbital ampullary organ field. Scale bars: 200 µm.

The online version of this article includes the following figure supplement(s) for figure 3:

**Figure supplement 1.** Examples of successful disruption of sterlet *Atoh1* by CRISPR/Cas9-mediated mutagenesis in G0-injected embryos.

**Figure supplement 2.** Examples of different phenotypes in *Atoh1* crispants.

itself scored as a phenotype. However, *Cacna1d* expression was absent mosaically in neuromast lines and ampullary organ fields in *Atoh1* crispants (*Figure 3E–J*; n=13/22, i.e., 59%, across five independent batches; *Supplementary file 1*). This suggested that disruption of the *Atoh1* gene in sterlet resulted in the failure of hair cell differentiation (as expected from zebrafish; *Millimaki et al., 2007*) and also electroreceptors. Post-ISH immunostaining for the supporting-cell marker Sox2 (*Hernández et al., 2007*; *Modrell et al., 2017a*; also see *Minařík et al., 2024*) confirmed that neuromasts had formed (*Figure 3E¹,F¹*; also see *Figure 3—figure supplement 2A–F*), so the phenotype was specific to receptor cells. Although post-ISH Sox2 immunostaining reliably revealed neuromasts, ampullary organs were only weakly and variably labelled by Sox2 immunostaining even before ISH in sterlet (*Figure 1A and B*; also *Minařík et al., 2024*), as in paddlefish (*Modrell et al., 2017a*). The signal was even weaker and more variable post-ISH, hence was only rarely detected in ampullary organs (see below).

ISH for electroreceptor-specific *Kcnab3* (*Modrell et al., 2017a*; also see *Minařík et al., 2024*) similarly showed no effect in *Tyr* control crispants (*Figure 3K–N*; n=0/34 across seven independent batches; *Supplementary file 1*), but the mosaic absence of *Kcnab3* expression in ampullary organ fields in *Atoh1* crispants (*Figure 3O–T*; n=8/13, i.e., 62%, across two independent batches; *Supplementary file 1*). Post-*Kcnab3* ISH immunostaining for Sox2, although very weak and variable in ampullary organs (versus reliable labelling of neuromasts), confirmed in a few cases that ampullary organs were still present, as well as neuromasts (*Figure 3O¹,P¹*, *Figure 3—figure supplement 2G–H¹*). Hence, the phenotype in ampullary organs was also specific to receptor cells. A schematic representation of the most severe *Atoh1* crispant phenotypes, compared to control *Tyr* crispants, is shown in *Figure 3U–X*.

In mouse cochlear hair cells, the 'hair cell' transcription factor genes *Pou4f3* and *Gfi1* are direct Atoh1 targets (*Yu et al., 2021*; *Jen et al., 2022*). ISH for *Pou4f3* showed no phenotype in *Tyr* controls (*Figure 4A and B*; n=0/9 across three batches; *Supplementary file 1*) but the mosaic absence of *Pou4f3* in ampullary organ fields and neuromast lines of *Atoh1* crispants (*Figure 4C–D¹*; n=13/15 larvae, i.e., 87%, across three batches; *Supplementary file 1*). Similarly, ISH for *Gfi1* showed no phenotype in *Tyr* controls (*Figure 4E and F*; n=0/17 across five batches; *Supplementary file 1*), but mosaic absence of *Gfi1* in ampullary organ fields and neuromast lines of *Atoh1* crispants (*Figure 4G–H¹*; n=9/14 larvae, i.e., 64%, across four batches; *Supplementary file 1*). Post-ISH immunostaining for Sox2 confirmed that neuromasts and ampullary organs were still present in *Atoh1* crispants (*Figure 4C¹,D¹,G¹,H¹*).

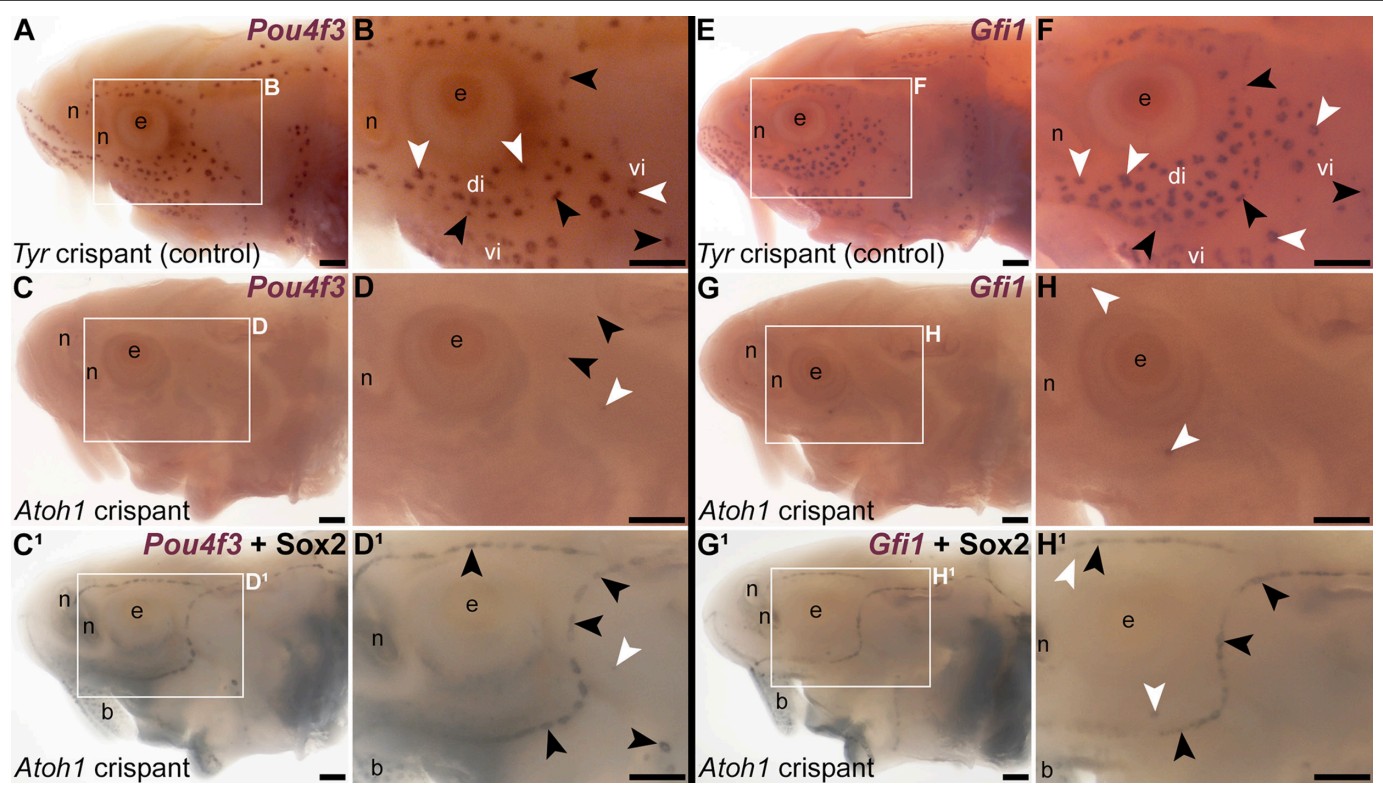

**Figure 4.** *Atoh1* is required for *Pou4f3* and *Gfi1* expression in ampullary organs and neuromasts. Sterlet crispants at stage 45 after in situ hybridisation (ISH) for transcription factor genes expressed by developing hair cells. Black arrowheads indicate examples of neuromasts; white arrowheads indicate examples of ampullary organs. (**A, B**) In a control *Tyr* crispant, *Pou4f3* expression is detected in both neuromasts and ampullary organs. (**C–D¹**) In an *Atoh1* crispant, *Pou4f3* expression is absent from both neuromasts and ampullary organs, except for a few isolated organs in the postorbital region. Post-ISH Sox2 immunostaining (**C¹, D¹**) shows that neuromast supporting cells are still present. Most ampullary organs are not visible as Sox2 expression in ampullary organs is significantly weaker than in neuromasts (**Figure 1A,B**) and often not detectable after post-ISH immunostaining. (**E, F**) In a control *Tyr* crispant, *Gfi1* expression is detected in both neuromasts and ampullary organs. (**G–H¹**) In an *Atoh1* crispant, *Gfi1* expression is absent from both neuromasts and ampullary organs, except for a few isolated organs in the supra- and infraorbital region. Post-ISH Sox2 immunostaining (**G¹, H¹**) shows that neuromast supporting cells are still present. (Most ampullary organs are not visible due to weaker Sox2 immunostaining.) Abbreviations: b, barbel; di, dorsal infraorbital ampullary organ field; e, eye; n, naris; vi, ventral infraorbital ampullary organ field. Scale bars: 200 μm.

Thus, these data suggest that *Pou4f3* and *Gfi1* are downstream of Atoh1 in ampullary organs as well as neuromasts.

Overall, across all riboprobes (*Cacna1d*, *Pou4f3*, and *Gfi1* labelling hair cells and electroreceptors; *Kcnab3* labelling electroreceptors only), around two-thirds of all *Atoh1* crispants analysed showed a mosaic absence of hair cells/electroreceptors (n=43/64, i.e., 67%; **Supplementary files 1** and **2**). In an attempt to analyse this phenotype more quantitatively, we scored all *Atoh1* crispants for severity of phenotype on the head in lateral view (**Supplementary file 2**). Approximately three-quarters of phenotypic *Atoh1* crispants had bilateral mosaic phenotypes (n=33/43, i.e., 77%; **Supplementary file 2**). The remaining crispants (n=10/43, i.e., 23%) had unilateral phenotypes. Furthermore, seven crispants with bilateral phenotypes were affected to different degrees on left and right sides of the head, hence 40% overall were differentially affected on left and right sides (n=17/43; **Supplementary file 2**), presumably as a result of holoblastic cleavage and CRISPR mosaicism. Given this, we scored the severity of the phenotype (mosaic absence of hair cells/electroreceptors) separately on each side of the head in lateral view. Ten of the 43 phenotypic crispants only had unilateral phenotypes, hence 76 out of 86 (88%) individual sides had a phenotype. Of these, fully 78% (n=59/76) were classed as 'severe', defined as more than two-thirds of cranial hair cells/electroreceptors absent in lateral view (examples shown in **Figure 3E,F,O,P**, **Figure 3—figure supplement 2J**). 11% (n=8/76) were classed as 'moderate' (between one-third and two-thirds of cranial hair cells/electroreceptors absent in lateral view; example shown in **Figure 3—figure supplement 2I**) and 12% (n=9/76) as 'mild' (less than

one-third of cranial hair cells/electroreceptors absent in lateral view; example shown in *Figure 3—figure supplement 2H*). Thus, the *Atoh1* sgRNA combination seemed to be relatively efficient and the phenotype, when present, was generally strong.

Taken together, these data suggest that Atoh1 is required for the differentiation of electroreceptors as well as hair cells and lies upstream of *Pou4f3* and *Gfi1* in ampullary organs as well as neuromasts.

## Targeting electrosensory-restricted *Neurod4* had no obvious effect on lateral line development

We previously identified *Neurod4* in paddlefish as the first-reported ampullary organ-restricted transcription factor in the developing lateral line system (*Modrell et al., 2017a*). We confirmed that sterlet *Neurod4* is similarly expressed by ampullary organs but not neuromasts (*Figure 1I and J*). We targeted *Neurod4* in sterlet embryos using nine different sgRNAs (*Table 1*, *Figure 1—figure supplement 1A*), injected in eight different combinations of 1–4 different sgRNAs across 10 independent batches of 1–2-cell-stage embryos (*Supplementary file 1*). This had no detectable effect on the expression of electroreceptor-specific *Kcnab3* (n=0/44 across nine batches, *Supplementary file 1*) or the hair cell/electroreceptor marker *Cacna1d* (n=0/4 across two batches, *Supplementary file 1*). Examples of *Neurod4* crispants after ISH for *Kcnab3*, plus a *Tyr* control crispant for comparison, are shown in *Figure 1—figure supplement 1B–D*. Our sgRNAs were designed before chromosome-level sterlet genomes were available (*Du et al., 2020* and the 2022 reference genome: GCF_902713425.1). Searching the reference genome for *Neurod4* showed that both ohnologs have been retained: one on chromosome 45 and the other annotated on an unplaced genomic scaffold, with 99.45% nucleotide identity (and 99.46% amino acid identity) in the coding sequence. Our sgRNAs target both *Neurod4* ohnologs without mismatches. Our genotyping primers were designed before chromosome-level sterlet genomes were available; comparison with the reference genome showed five mismatches in the forward primer used for genotyping larvae targeted with sgRNAs 1 and 2 against the chromosome 45 ohnolog (*Supplementary file 1*). It was not possible to tell from our genotyping data whether the primers amplified both ohnologs or only one, however, as the remaining sequence targeted by the primers is identical between the two ohnologs. Genotyping and ICE analysis (*Conant et al., 2022*) showed successful disruption of the *Neurod4* gene arising from six different combinations of the sgRNAs (*Supplementary file 1*; examples are shown in *Figure 1—figure supplement 1E–I*). Not all sgRNA combinations were genotyped, but all *Neurod4* crispants counted (n=48) included at least one sgRNA confirmed to disrupt the *Neurod4* gene via genotyping of other larvae (*Supplementary file 1*).

The lack of phenotype in *Neurod4* crispants (n=0/48 across all markers; *Supplementary file 1*), despite successful disruption of the *Neurod4* gene, suggested either that Neurod4 is not required for electroreceptor differentiation, despite its restriction to ampullary organs in both paddlefish and sterlet (*Modrell et al., 2017a*; this paper), or that it acts redundantly with other transcription factors. In paddlefish, *Neurod1* expression was restricted to developing lateral line ganglia (*Modrell et al., 2011b*). We cloned sterlet *Neurod1*, *Neurod2,* and *Neurod6*. (Unlike *Neurod4*, these three *Neurod* family members are all direct Atoh1 targets in mouse hair cells; *Jen et al., 2022*.) *Neurod1* and *Neurod6* proved to be expressed in sterlet ampullary organs, as well as neuromasts, while *Neurod2* was not expressed in either (*Figure 1—figure supplement 2*). Thus, it seems likely that the lack of effect of CRISPR/Cas9-mediated targeting of sterlet *Neurod4* is due to redundancy with other Neurod family transcription factors co-expressed in ampullary organs. Our results also show there is at least some variation in *Neurod* family gene expression within the developing lateral line systems of paddlefish and sterlet.

## Targeting mechanosensory-restricted *Foxg1* led to ectopic ampullary organs forming within neuromast lines and gaps where neuromasts are missing

We recently identified *Foxg1* as a mechanosensory lateral line-restricted transcription factor gene in paddlefish and sterlet (*Minařík et al., 2024*). *Foxg1* is expressed in the central zones of lateral line sensory ridges where neuromasts form, though excluded from hair cells themselves (*Minařík et al., 2024*). Our sgRNAs against *Foxg1* (*Table 1*, *Figure 5—figure supplement 1A*) were designed before chromosome-level sterlet genomes were available (*Du et al., 2020* and the 2022 reference genome GCF_902713425.1). Searching the reference genome for *Foxg1* showed that both *Foxg1* ohnologs

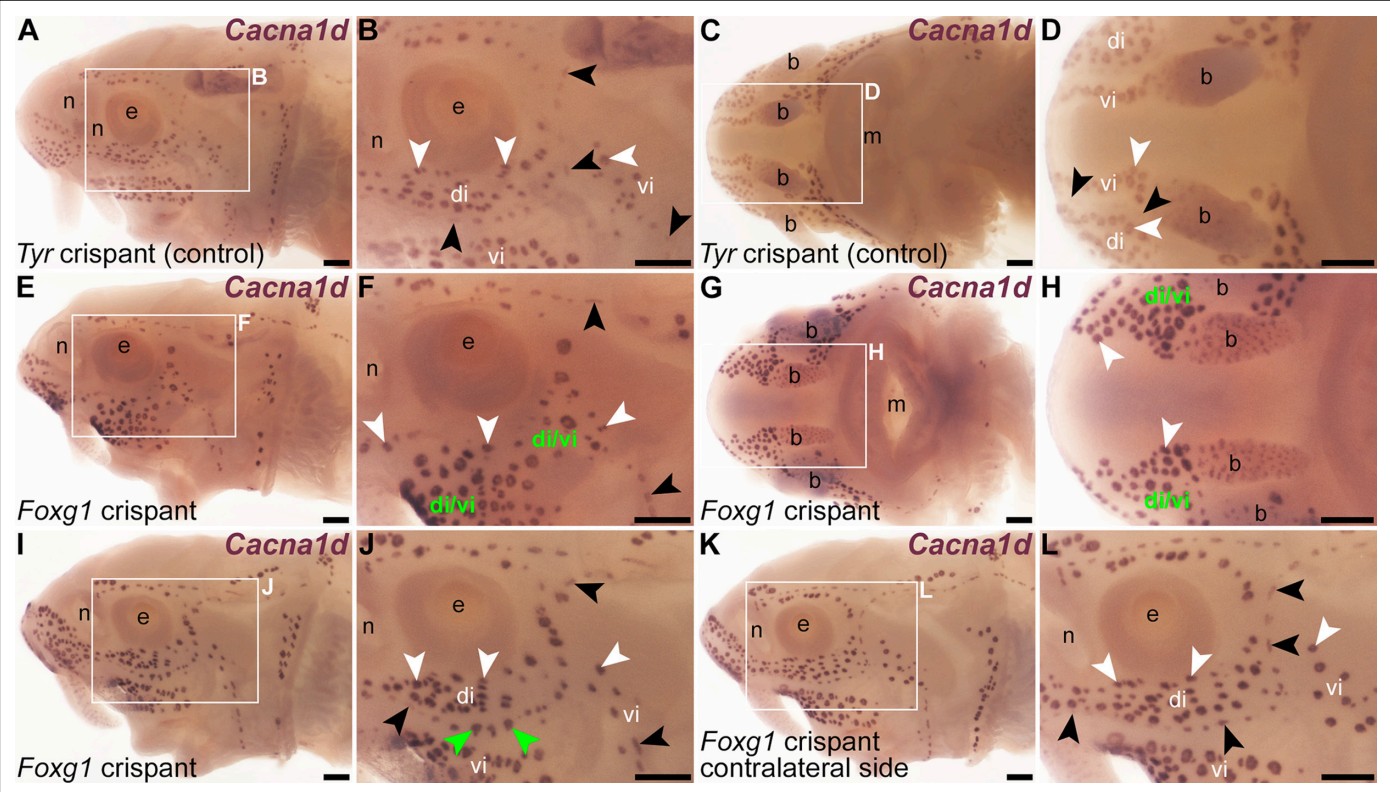

**Figure 5.** Neuromast lines in *Foxg1* crispants are disrupted by putative ampullary organs. Sterlet crispants at stage 45 after in situ hybridisation (ISH) for the hair cell and electroreceptor marker *Cacna1d*. Black arrowheads indicate examples of neuromasts; white arrowheads indicate examples of ampullary organs. (**A, B**) Lateral view of a control *Tyr* crispant. *Cacna1d* expression shows the normal distribution of hair cells and electroreceptors. Note that ampullary organs have significantly more *Cacna1d*-expressing receptor cells than neuromasts. (**C, D**) Ventral view of a second control *Tyr* crispant. *Cacna1d* expression reveals the infraorbital neuromast line on both sides of the ventral rostrum, flanked by the dorsal infraorbital (di) and ventral infraorbital (vi) ampullary organ fields. (**E, F**) Lateral view of a *Foxg1* crispant. *Cacna1d* expression reveals that distinct neuromast lines are missing and the corresponding space is filled by putative ectopic ampullary organs, based on the large, ampullary organ-like clusters of *Cacna1d*-expressing cells. The dorsal and ventral infraorbital ampullary organ fields seem to have fused across the missing neuromast line (compare with **A, B**). (**G, H**) Ventral view of a second *Foxg1* crispant. *Cacna1d* expression reveals an apparent fusion of the dorsal infraorbital (di) and ventral infraorbital (vi) ampullary organ fields across the missing infraorbital neuromast lines on both sides (compare with **C, D**). (**I–L**) In a third *Foxg1* crispant, *Cacna1d* expression on the left side (**I, J**) shows that distinct supraorbital and infraorbital neuromast lines are still present. However, some organs within the supraorbital line and most organs within the infraorbital line have large clusters of *Cacna1d*-expressing cells, suggesting they are ectopic ampullary organs (green arrowheads in panel **J** show examples). On the right side (**K, L**; image flipped horizontally for ease of comparison), this phenotype is not seen. Abbreviations: b, barbel; di, dorsal infraorbital ampullary organ field; di/vi, fused dorsal infraorbital and ventral infraorbital ampullary organ fields; e, eye; m, mouth; n, naris; vi, ventral infraorbital ampullary organ field. Scale bars: 200 μm.

The online version of this article includes the following figure supplement(s) for figure 5:

**Figure supplement 1.** Examples of successful disruption of sterlet *Foxg1* by CRISPR/Cas9-mediated mutagenesis in G0-injected embryos.

have been retained, on chromosomes 15 and 18, with 96.43% nucleotide identity in the coding sequence (99.51% amino acid identity). Our sgRNAs target both *Foxg1* ohnologs without mismatches. Although our genotyping primers were designed before chromosome-level sterlet genomes were available, comparison with the reference genome showed no mismatches against either of the two ohnologs. Genotyping and ICE analysis (*Conant et al., 2022*) of tails from individual *Foxg1* crispants showed successful disruption of the *Foxg1* gene (*Figure 5—figure supplement 1* shows examples; the genotyping data were consistent with the primers amplifying both ohnologs).

When compared with *Tyr* control crispants (*Figure 5A–D*), a striking phenotype was seen mosaically after targeting mechanosensory-restricted *Foxg1* with sgRNAs 1 and 2 (*Table 1*, *Figure 5—figure supplement 1*): neuromast lines were often interrupted by ectopic ampullary organs. These were defined as *Cacna1d*-expressing cells present within neuromast lines in larger clusters than expected for neuromasts, resembling ampullary organs (*Figure 5E–L*; n=8/18, i.e., 44%, across two independent

batches; *Supplementary file 1*), or by expression of electroreceptor-specific *Kcnab3* within neuromast lines, not seen in *Tyr* control crispants (*Figure 6A–P*; n=15/24, i.e., 63%, across both batches; *Supplementary file 1*). This sometimes led to the apparent fusion of ampullary organ fields across a missing neuromast line (for example, *Figure 5E–H* and *Figure 6O, P*). Post-ISH Sox2 immunostaining of selected *Foxg1* crispants showed that some phenotypes were initially missed, as the electroreceptor-specific *Kcnab3* labelling did not provide sufficient topographical context to assess ampullary organ field expansion into the infraorbital (*Figure 6F*) and preopercular neuromast lines (*Figure 6L*). Post-*Kcnab3* ISH immunostaining for Sox2 was thus performed in all crispants to reveal neuromasts; this also confirmed that the ectopic electroreceptors in *Foxg1* crispants were located within neuromast lines (*Figure 6B–D,F,H,J,L*). Targeting *Foxg1* with a different pair of sgRNAs (*Table 1*, *Figure 5—figure supplement 1A*) in a different batch of embryos generated similar phenotypes, that is, the interruption of neuromast lines by ectopic ampullary organs, in 7/21 crispants (33%) overall (*Cacna1d*: n=0/6; *Kcnab3*: n=7/15; *Supplementary file 1*). We also investigated this phenotype by performing ISH for two ampullary organ-restricted transcription factor genes, *Mafa* (*Minařík et al., 2024*) and *Neurod4* (*Modrell et al., 2017a*; this study), followed by Sox2 immunostaining to reveal the location of neuromasts. Relative to the normal ampullary organ expression of *Mafa* in uninjected siblings/half-siblings (sufficient *Tyr* control crispants were not available to test; *Figure 7A-D*; n=0/6 larvae within one batch; *Supplementary file 1*), ISH for *Mafa* showed the mosaic presence of ampullary organs in neuromast lines and/or merging of ampullary organ fields in *Foxg1* crispants (*Figure 7E–J*; n=9/18 crispants, i.e., 50%, across two independent batches; *Supplementary file 1*). Similarly, relative to the usual ampullary organ expression of *Neurod4* in uninjected siblings/half-siblings (sufficient *Tyr* control crispants were not available to test; *Figure 7K–N*; n=0/2 larvae within one batch; *Supplementary file 1*), ISH for *Neurod4* (more weakly expressed than *Mafa*) revealed the same ectopic ampullary organ phenotype in *Foxg1* crispants as seen for *Cacna1d*, *Kcnab3*, and *Mafa* (*Figure 7O–T*; n=3/6 crispants, i.e., 50%, within one batch; *Supplementary file 1*).

Post-ISH Sox2 immunostaining of all *Foxg1* crispants, together with skin-mount analysis, enabled us to study the disruption of neuromast lines by ectopic ampullary organs in more detail (*Figure 7—figure supplement 1*). Strikingly, in several cases where small patches of *Kcnab3* or *Mafa*-positive electroreceptors seemingly developed within uninterrupted neuromast lines, skin-mount analysis after Sox2 immunostaining showed that ectopic electroreceptors had formed within otherwise *Kcnab3* or *Mafa*-negative neuromasts, that is, with the expected elongated morphology and stronger post-ISH Sox2 immunoreactivity of neuromasts (*Figure 8A–J*; n=7). This suggests that *Foxg1* also represses electroreceptor formation within developing neuromasts.

Furthermore, post-ISH Sox2 immunostaining revealed gaps in cranial neuromast lines in around half of all *Foxg1* crispants (across all riboprobes) that were not interrupted by ectopic ampullary organs (n=48/87, i.e., 55%; *Supplementary file 3*). This was most commonly seen within the otic neuromast line (for example, compare the otic neuromast line in the *Tyr* control in *Figure 6I* with that of the *Foxg1* crispant shown in *Figure 6J*), but also in the infraorbital and supraorbital lines (for example, compare *Figure 6G and H*, *Figure 6E and F*). This 'missing neuromasts' phenotype was not seen in *Tyr* control crispants (n=0/20).

Overall, 64% (n=56/87) of *Foxg1* crispants showed a phenotype (*Supplementary file 3*). In the phenotypic *Foxg1* crispants, neuromast lines were interrupted by ectopic ampullary organs in 75% (n=42/56), by gaps in 86% (n=48/56) and by both in 61% (n=34/56) (*Supplementary file 3*). Electroreceptors were identified within neuromasts in 13% of phenotypic *Foxg1* crispants (n=7/56; *Supplementary file 3*). In an attempt to quantify the observed phenotypes further, we scored all 56 phenotypic *Foxg1* crispants for the degree of disruption to pre-otic neuromast lines in both lateral and ventral view (so as to score the entirety of the infraorbital line), whether by the interruption of neuromast lines by ectopic ampullary organs and/or missing neuromasts (*Supplementary file 3*). Most phenotypic crispants were differentially affected on left versus right sides of the head, with different neuromast lines affected on each side (n=50/56; 89%) including entirely unilateral phenotypes (n=38/56, 68%; *Supplementary file 3*), so we scored each side separately (112 sides). We assessed each of the pre-otic neuromast lines separately (see schematic in *Figure 9A*): the supraorbital and infraorbital lines (both of which derive from the anterodorsal lateral line placode; *Gibbs and Northcutt, 2004*), the otic line (derived from the otic lateral line placode; *Gibbs and Northcutt, 2004*), and the preopercular line (derived from the anteroventral lateral line placode; *Gibbs and Northcutt, 2004*). We ignored

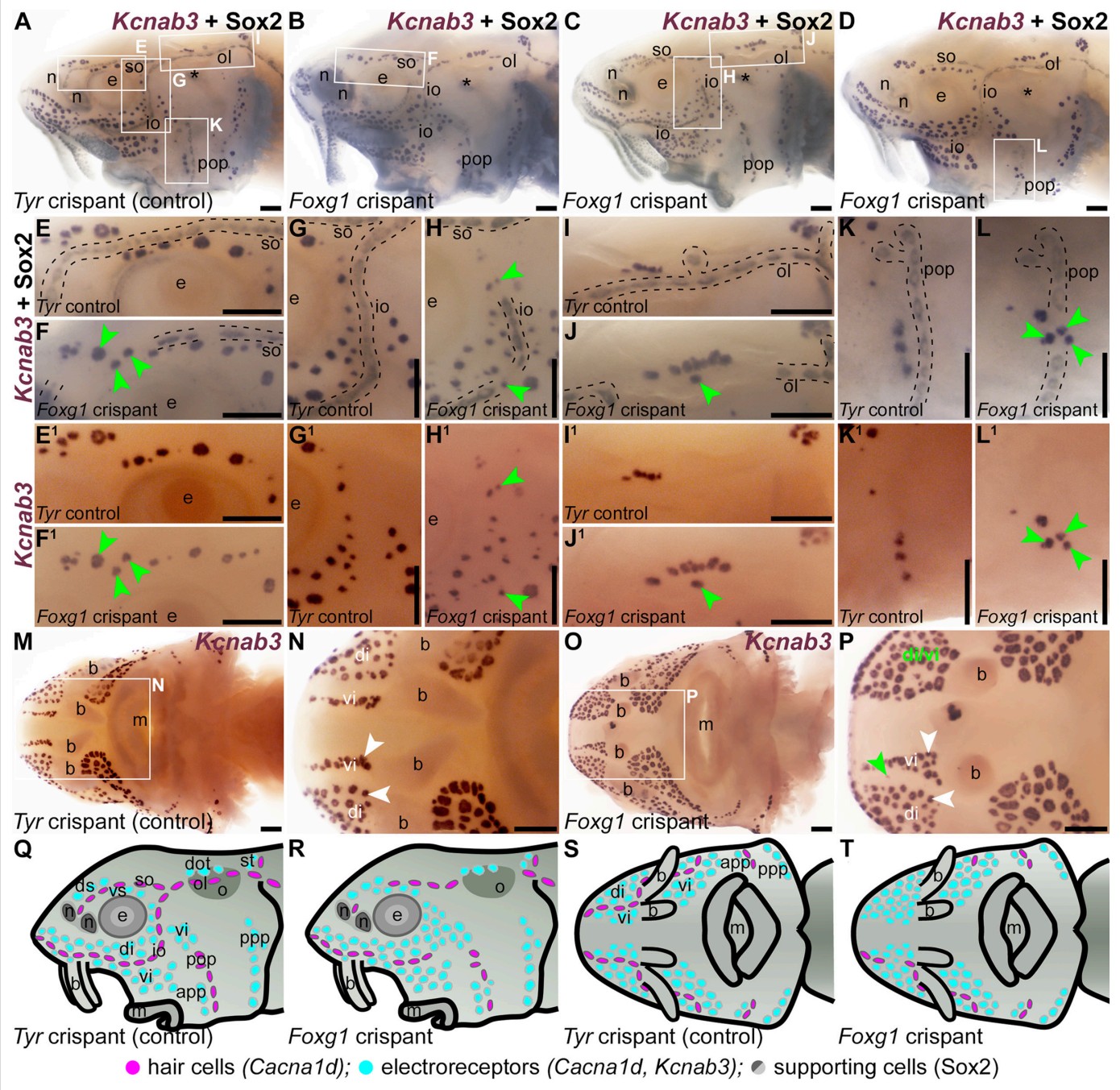

**Figure 6.** Neuromast lines in *Foxg1* crispants are disrupted by ectopic ampullary organs and missing neuromasts. Sterlet crispants at stage 45 after in situ hybridisation (ISH) for the electroreceptor-specific marker *Kcnab3*. White arrowheads indicate examples of ampullary organs. (**A–D**) A control *Tyr* crispant (**A**) and three *Foxg1* crispants (**B–D**) after post-ISH Sox2 immunostaining. Black asterisk indicates the spiracular opening (first pharyngeal cleft). (**E–L**) Higher-power views of individual neuromast lines (outlined with dashed lines) from the *Tyr* control (**E, G ,I ,K**) and *Foxg1* crispants (**F, H, J, L**) after post-ISH Sox2 immunostaining (locations indicated by boxes on panels **A–D**). Comparing the *Tyr* control and *Foxg1* crispants shows ectopic ampullary organs (green arrowheads) and gaps (where neuromasts are missing) disrupting the supraorbital line (**E, F**), infraorbital line (**G, H**), and otic line (**I, J**), and ectopic ampullary organs disrupting the preopercular line (**K, L**). (**E¹–L¹**) The same areas shown before Sox2 immunostaining. Electroreceptor-specific *Kcnab3* expression shows the distribution of ampullary organs only. (**M, N**) Ventral view of the same control *Tyr* crispant as in panel **A**, before Sox2 immunostaining. *Kcnab3* expression shows the distribution of ampullary organ fields. Note the lack of staining where the infraorbital neuromast lines run on either side of the ventral rostrum, flanked by the dorsal infraorbital (di) and ventral infraorbital (vi) ampullary organ fields (compare with *Cacna1d* expression in **Figure 2C,D**). (**O, P**) Ventral view of a fourth *Foxg1* crispant. On the left side, ectopic *Kcnab3*-expressing electroreceptors fill the space where the left infraorbital neuromast line would normally run, such that the dorsal and ventral infraorbital ampullary organ fields seem to have

*Figure 6 continued on next page*

*Figure 6 continued*

fused (compare with **M, N**). (**Q–T**) Schematic representation of cranial lateral line organs in a stage 45 control *Tyr* crispant (lateral view, **Q**; ventral view, **S**) versus a *Foxg1* crispant in which the pre-otic neuromast lines are disrupted by ectopic ampullary organs or, as shown for the otic line, gaps where neuromasts are missing (lateral view, **R**; ventral view, **T**). Abbreviations: app, anterior preopercular ampullary organ field; b, barbel; di, dorsal infraorbital ampullary organ field; di/vi, fused dorsal infraorbital and ventral infraorbital ampullary organ fields; dot, dorsal otic ampullary organ field; ds, dorsal supraorbital ampullary organ field; e, eye; io, infraorbital neuromast line; m, mouth; n, naris; o, otic capsule; ol, otic neuromast line; pop, preopercular neuromast line; ppp, posterior preopercular ampullary organ field; so, supraorbital neuromast line; st, supratemporal ampullary organ field; vi, ventral infraorbital ampullary organ field; vs, ventral supraorbital ampullary organ field. Scale bars: 200 μm.

the post-otic supratemporal neuromast line, which is very short, and the middle and posterior neuromast lines, which are not associated with ampullary organ fields (see schematic in *Figure 9A*). The phenotype was classed as 'severe' when more than two-thirds of the neuromast line was affected; 'moderate' when between one-third and two-thirds of the line was affected; and 'mild' when less than one-third of the line was affected.

At least one neuromast line was classed as severely disrupted in 38% of the 56 phenotypic crispants (n=21/56; *Supplementary file 3*). The maximum degree of severity was scored as 'moderate' for 32% (n=18/56) and 'mild' for the remaining 30% (n=17/56) (*Supplementary file 3*). In total, 112 individual sides (from 56 phenotypic *Foxg1* crispants) were scored, of which 66% (n=74/112) had at least one disrupted neuromast line. *Supplementary file 3* shows the scoring information for the supraorbital, infraorbital, otic, and preopercular neuromast lines on each side of each individual *Foxg1* crispant, plus summary data for each neuromast line across those *Foxg1* crispants with a phenotype as described below. The summary data are also presented in bar charts in *Figure 9B–D*.

The most commonly disrupted neuromast line was the otic line (76%; n=56/74 phenotypic sides of *Foxg1* crispants; *Figure 9B*), with a severe phenotype scored in 29% of cases (n=16/56; *Figure 9C*). In all but one phenotypic side, the otic line phenotype included gaps (n=55/56 disrupted otic lines, i.e., 98%; *Figure 9D*) and, in most cases, was due to gaps alone (n=49/56, i.e., 88%; *Figure 9D*). Ectopic ampullary organs were only seen in 11% of disrupted otic lines (n=6/56; *Figure 9D*). Electroreceptors were identified inside neuromasts in a single disrupted otic line (2%; n=1/56). Conversely, the preopercular neuromast line was the least commonly affected (26%; n=19/74 phenotypic sides of *Foxg1* crispants; *Figure 9B*) and the phenotype was always mild (i.e., less than one-third of the line was affected; n=19/19; *Figure 9C*). Also in contrast to the otic line, the preopercular line phenotype almost always included ectopic ampullary organs (n=18/19, i.e., 95%; *Figure 9D*) and, in most cases, was due to ectopic ampullary organs alone (n=15/19, i.e., 79%; *Figure 9D*). Additional gaps were only seen in 21% of disrupted preopercular lines (n=4/19; *Figure 9D*) and only one preopercular line was disrupted by a gap alone (5%; n=1/19; *Figure 9D*). The phenotypes seen in the supraorbital and infraorbital neuromast lines fell between these extremes, with the infraorbital affected a little more than the supraorbital. The infraorbital line was disrupted in 47% of phenotypic *Foxg1* crispant sides (n=35/74; *Figure 9B*), with a severe phenotype in 37% of cases (n=13/35; *Figure 9C*). The supraorbital line was affected in 36% of phenotypic *Foxg1* crispant sides (n=27/74; *Figure 9B*), with a severe phenotype in only 19% of cases (n=5/27; *Figure 9C*). For both lines, the phenotype usually included ectopic ampullary organs (infraorbital: n=26/35, i.e., 74%; supraorbital: n=23/27, i.e., 85%; *Figure 9D*) and often included gaps (infraorbital: n=17/35, i.e., 49%; supraorbital: n=17/27, i.e., 63%; *Figure 9D*). Electroreceptors were identified inside neuromasts in 20% of disrupted infraorbital lines (n=7/35); in one of these crispants, both the infraorbital and otic lines on the right side of the head showed this phenotype (the only otic line with this phenotype, mentioned above). Ectopic ampullary organs alone were seen in 31% of disrupted infraorbital lines (n=11/35) and 37% of disrupted supraorbital lines (n=10/27). Gaps alone were only seen in 17% of disrupted infraorbital lines (n=6/35; *Figure 9D*) and 15% of disrupted supraorbital lines (n=4/27; *Figure 9D*).

Overall, the scoring data suggest a loose relationship between the type of disruption seen in the neuromast lines of *Foxg1* crispants (*Supplementary file 3*; *Figure 9B–D*) and the size of the associated ampullary organ fields (see schematic in *Figure 9A*). The otic neuromast line, which is associated with only the small dorsal otic ampullary organ field (and in which the first neuromasts differentiate: see *Minařík et al., 2024*), was the most commonly disrupted, but in almost all cases by gaps alone, without ectopic ampullary organs (*Figure 9B,D*). The other three lines, which are all flanked by larger

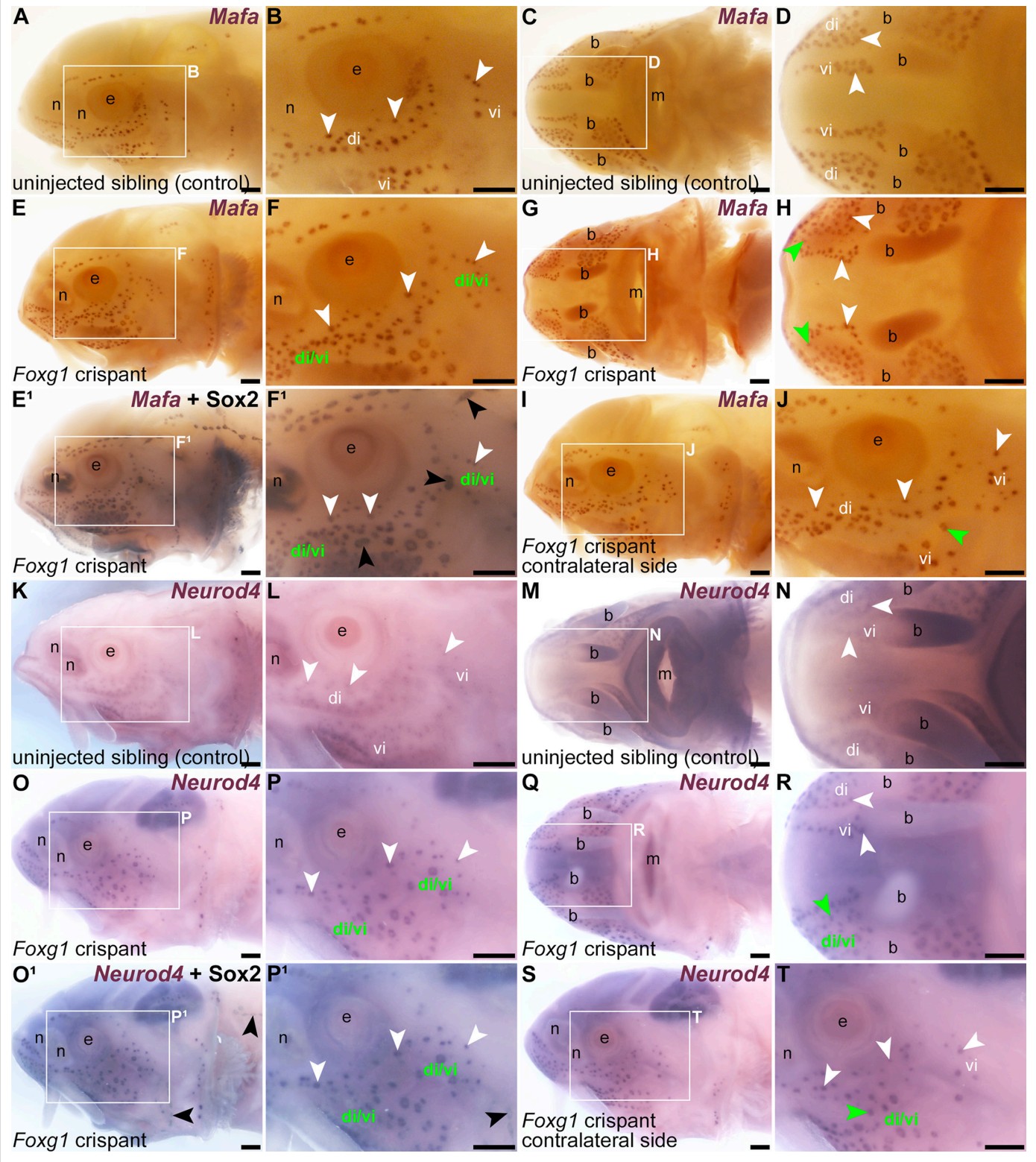

**Figure 7.** Ectopic ampullary organs in *Foxg1* crispants express ampullary organ-specific transcription factor genes *Mafa* and *Neurod4*. Sterlet crispants at stage 45 after in situ hybridisation (ISH) for ampullary organ-restricted transcription factor genes. Black arrowheads indicate examples of neuromasts; white arrowheads indicate examples of ampullary organs. (**A–D**) In an uninjected sibling/half-sibling (eggs were fertilised in vitro with a mix of sperm from three different males), *Mafa* expression is restricted to ampullary organs (lateral view: **A, B**; ventral view: **C, D**). (**E–J**) A *Foxg1* crispant. On the left side of the head (**E, F**), several *Mafa*-expressing ectopic ampullary organs are present in the space where the infraorbital neuromast line would

*Figure 7 continued on next page*

*Figure 7 continued*

normally run, such that the dorsal and ventral infraorbital ampullary organ fields seem to have fused (compare with **A, B**). Post-ISH Sox2 immunostaining (**E[1], F[1]**) shows that neuromasts are still present both proximally and distally to the sites of ampullary organ field fusion. In ventral view (**G, H**), ectopic ampullary organs (green arrowheads) are seen bilaterally, within the spaces where the infraorbital neuromast lines run on either side of the ventral rostrum (compare with **C, D**). On the right side in lateral view (**I, J**; image flipped horizontally for ease of comparison), a single *Mafa*-expressing ectopic ampullary organ (green arrowhead) is also present in the space where the infraorbital neuromast line runs (compare with **A, B**). (**K–N**) In an uninjected sibling/half-sibling, *Neurod4* expression is restricted to ampullary organs (lateral view: **K, L**; ventral view: **M, N**). (**O–T**) A *Foxg1* crispant. On the left side of the head (**O, P**), *Neurod4*-expressing ectopic ampullary organs are present in the space where the infraorbital neuromast line would normally run, such that the dorsal and ventral infraorbital ampullary organ fields seem to have fused (compare with **K, L**). Post-ISH Sox2 immunostaining (**O[1], P[1]**) suggests that neuromasts are absent from the site of infraorbital ampullary organ field fusion, although neuromasts can be seen in the preopercular and trunk lines (black arrowheads, compare with **O, P**). In ventral view (**Q, R**), ectopic *Neurod4*-expressing ampullary organs are seen where the right infraorbital neuromast line would normally run on the ventral rostrum (green arrowhead indicates an example), resulting in partial fusion of the dorsal and ventral infraorbital fields on this side (the left side is unaffected). On the right side in lateral view (**S, T**; image flipped horizontally for ease of comparison), ectopic ampullary organs are also present in the space where the infraorbital neuromast line runs (green arrowhead in panel **T** indicates an example), resulting in the apparent partial fusion of the dorsal and ventral infraorbital ampullary organ fields (compare with **K, L**). Abbreviations: b, barbel; di, dorsal infraorbital ampullary organ field; di/vi, fused dorsal infraorbital and ventral infraorbital ampullary organ fields; e, eye; m, mouth; n, naris; vi, ventral infraorbital ampullary organ field. Scale bars: 200 µm.

The online version of this article includes the following figure supplement(s) for figure 7:

**Figure supplement 1.** *Foxg1* crispant phenotypes include ectopic ampullary organs within neuromast lines.

ampullary organ fields on both sides, were mostly (infraorbital and supraorbital lines) or almost always (preopercular line) disrupted by ectopic ampullary organs, as well as by gaps (*Figure 9B,D*).

Taken together, these data suggest that mechanosensory-restricted FoxG1 is necessary for the formation and/or maintenance of neuromasts (as recently reported in zebrafish; *Bell et al., 2024*) and also acts (whether directly or indirectly) to repress the formation of ampullary organs and electroreceptors within neuromast lines and electroreceptors within neuromasts.

## Discussion
### Conserved molecular mechanisms underlie lateral line electroreceptor and hair cell formation

Here, we aimed to test the function in lateral line electroreceptor and/or hair cell formation of three transcription factor genes that we had previously identified as expressed in developing electrosensory ampullary organs and/or mechanosensory lateral line neuromasts in ray-finned chondrostean fishes—paddlefish and sterlet (*Butts et al., 2014*; *Modrell et al., 2017a*; *Minařík et al., 2024*). The first gene we investigated was *Atoh1*, which is required for the formation of lateral line hair cells in zebrafish (*Millimaki et al., 2007*), as well as hair cells in the inner ear (*Bermingham et al., 1999*; *Millimaki et al., 2007*). In paddlefish (*Butts et al., 2014*; *Modrell et al., 2017a*) and sterlet (*Minařík et al., 2024*), *Atoh1* is expressed in ampullary organs as well as neuromasts. Targeting *Atoh1* for CRISPR/Cas9-mediated mutagenesis in G0 sterlet embryos showed that Atoh1 is required for the formation not only of *Cacna1d*-expressing neuromast hair cells, as expected from zebrafish (*Millimaki et al., 2007*), but also of *Cacna1d*-expressing, *Kcnab3*-expressing electroreceptors. These experiments also showed that Atoh1 is required for the expression of the 'hair cell' transcription factor genes *Gfi1* and *Pou4f3* in developing ampullary organs, as well as neuromasts. This is consistent with both of these genes being direct Atoh1 targets in mouse cochlear hair cells (*Yu et al., 2021*; *Jen et al., 2022*).

In both inner-ear hair cells and Merkel cells (epidermal mechanoreceptors found in all vertebrates; see, for example, *Whitear, 1989*; *Brown et al., 2023*), Atoh1 acts with Pou4f3 in a conserved 'feed-forward circuit', with Atoh1 directly activating *Pou4f3* expression, and Pou4f3 then acting as a pioneer factor to open a significant subset of Atoh1 target enhancers (some shared and some divergent between hair cells and Merkel cells), enabling mechanosensory differentiation (*Yu et al., 2021*). In hair cells, *Gfi1* is one of the Pou4f3-dependent Atoh1 targets (*Yu et al., 2021*). Together with the striking conservation of transcription factor gene expression between developing ampullary organs and neuromasts (*Modrell et al., 2011a*; *Modrell et al., 2011b*; *Modrell et al., 2017a*; *Modrell et al., 2017b*; also *Minařík et al., 2024*), the phenotypes seen in *Atoh1*-targeted G0 sterlet crispants suggest that the molecular mechanisms underlying electroreceptor formation are highly conserved

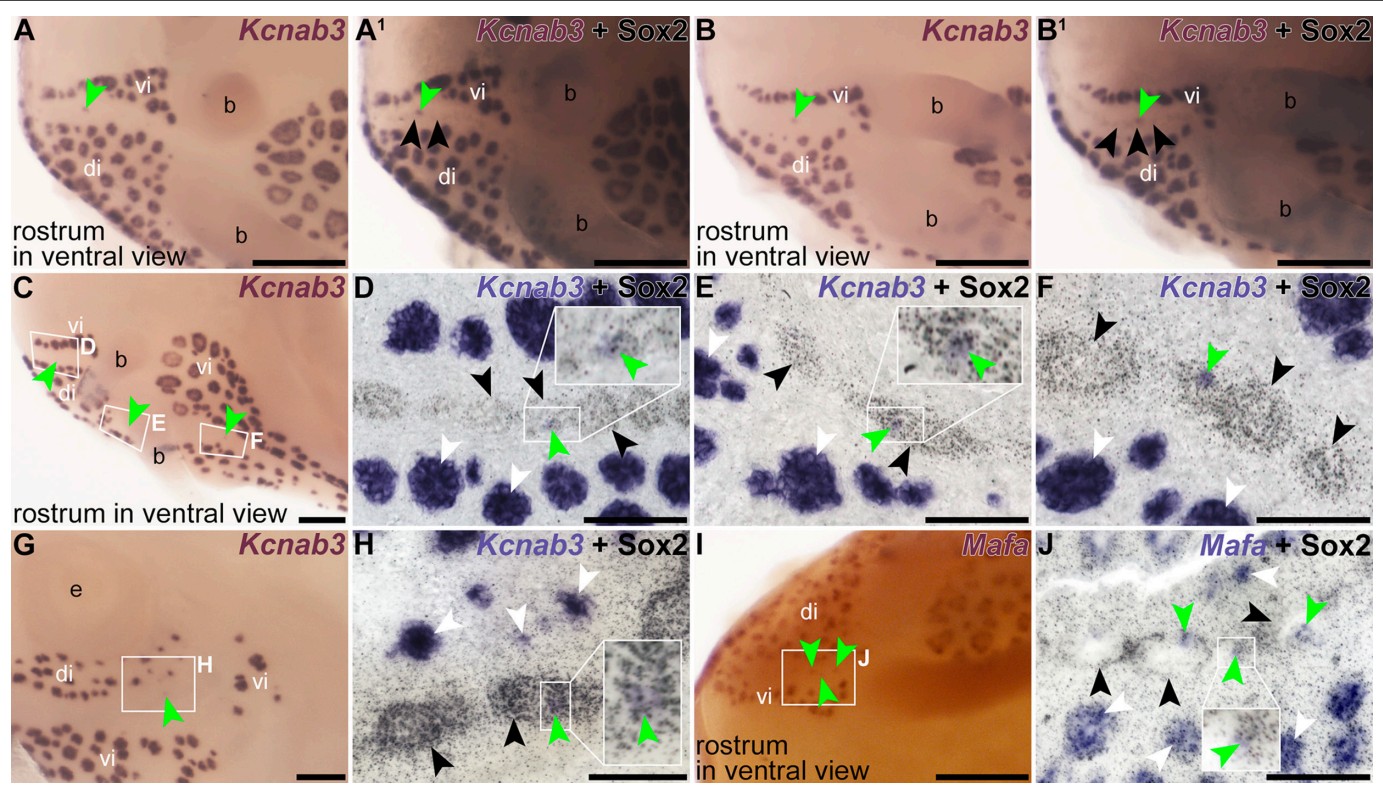

**Figure 8.** *Foxg1* crispant phenotypes include the presence of ectopic electroreceptors within individual neuromasts. Sterlet *Foxg1* crispants at stage 45 after in situ hybridisation (ISH) for electroreceptor-specific markers. All images show a ventral view of the right side of the rostrum (compare with schematic in *Figure 6S,T*) except panel G (lateral view). Black arrowheads indicate examples of neuromasts; white arrowheads indicate examples of ampullary organs; green arrowheads indicate ectopic electroreceptors. (**A–B**[1]) Two *Foxg1* crispants in which a small patch of electroreceptor-specific *Kcnab3* expression is seen in the apparent path of the right infraorbital neuromast line on the ventral rostrum (compare with the large clusters of electroreceptors in the ampullary organs on either side, in the ventral infraorbital and dorsal infraorbital fields). Post-ISH Sox2 immunostaining (**B, B**[1]) reveals that the neuromast line remains uninterrupted, suggesting that the ectopic electroreceptors formed within neuromasts. (**C–F**) A third *Foxg1* crispant after ISH for *Kcnab3*. In wholemount (**C**), a tiny patch of electroreceptor-specific *Kcnab3* expression is visible in the apparent path of the right infraorbital neuromast line on the ventral rostrum (green arrowhead in boxed region labelled **F**). The other two green arrowheads indicate the positions of even smaller patches of *Kcnab3* expression, too faint to be visible in wholemount images. Skin mounts from this region, imaged after post-ISH Sox2 immunostaining (**D–F**; locations shown in boxed regions on panel **C**), show that all of the ectopic *Kcnab3*-positive electroreceptors are located within Sox2-positive neuromasts in an uninterrupted neuromast line. (**G–J**) The same phenotype can be observed in a fourth *Foxg1* crispant after ISH for *Kcnab3* (**G, H**) and a fifth *Foxg1* crispant after ISH for the electroreceptor-specific marker *Mafa* (**I,J**). Abbreviations: b, barbel; di, dorsal infraorbital ampullary organ field; vi, ventral infraorbital ampullary organ field. Scale bars: **A–C, G, I**, 200 μm; **D–F, H, J**, 50 μm.

with those underlying hair cell formation. Indeed, the requirement of Atoh1 for *Pou4f3* and *Gfi1* expression in ampullary organs, as well as neuromasts, suggests that the Atoh1-Pou4f3 'feed-forward circuit' in mechanosensory cells—that is, hair cells and epidermal Merkel cells (*Yu et al., 2021*)—may also be conserved, at least partly, in developing electroreceptors. Taken together, these data support the hypothesis that electroreceptors evolved as a transcriptionally related 'sister cell type' to lateral line hair cells (*Arendt et al., 2016*; *Baker and Modrell, 2018*; *Baker, 2019*).

## Electrosensory-restricted Neurod4 is likely redundant with other Neurod family members in sterlet

Paddlefish *Neurod4* was the first-reported ampullary organ-restricted transcription factor gene (*Modrell et al., 2017a*), with conserved expression in sterlet (this study). We were unable to detect a lateral line organ phenotype in *Neurod4*-targeted sterlet crispants. However, we found that *Neurod1* and *Neurod6* (but not *Neurod2*) are also expressed in sterlet ampullary organs (as well as neuro-masts), suggesting that Neurod4 may act redundantly with one or both of these factors in developing ampullary organs. (In paddlefish, however, *Neurod1* expression is restricted to developing lateral line

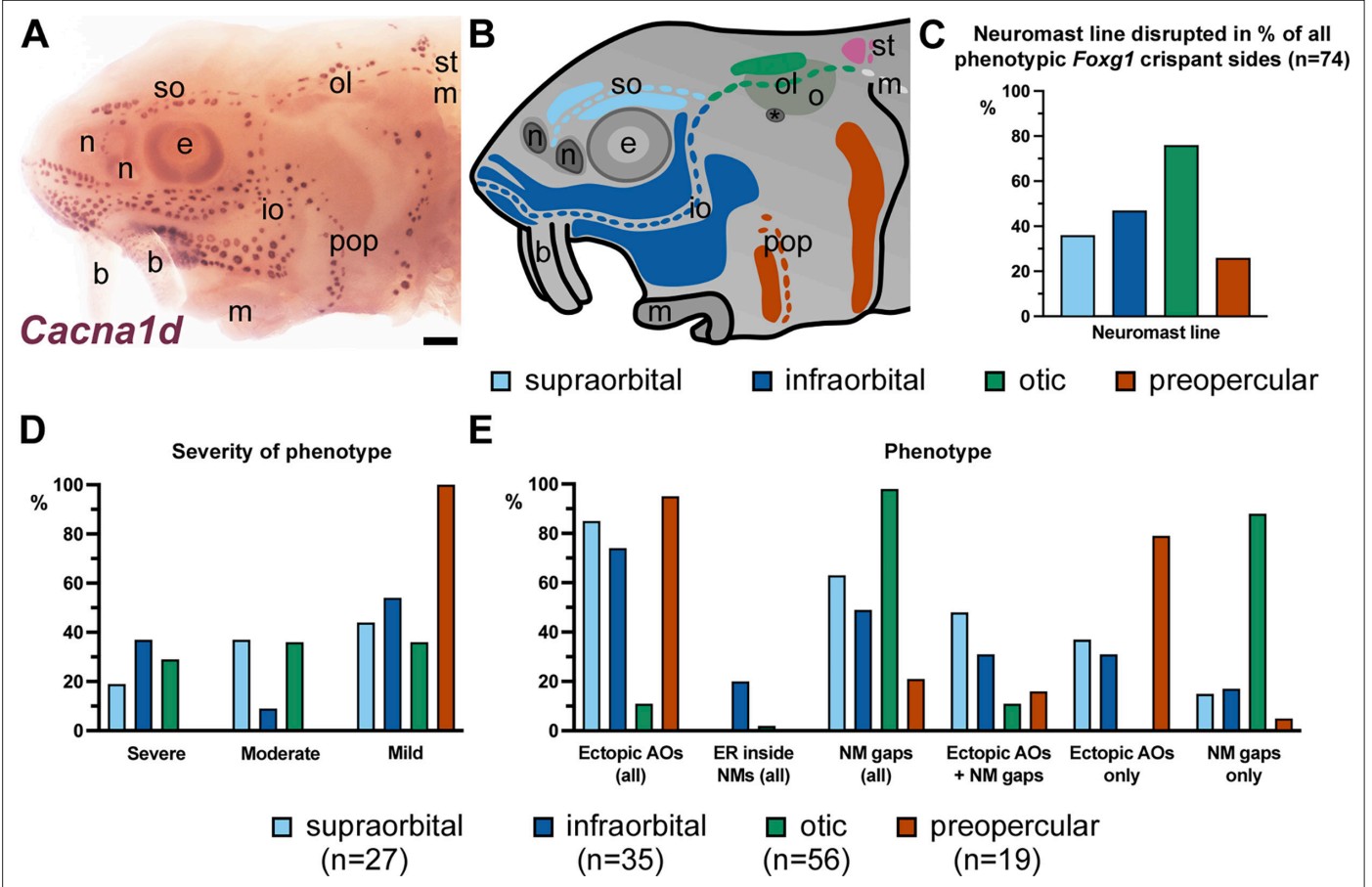

**Figure 9.** Differential disruption of individual neuromast lines in *Foxg1* crispants. (**A**) In situ hybridisation for the hair cell/electroreceptor marker *Cacna1d* at stage 45 identifies neuromasts and ampullary organs. (**B**) Schematic of a stage 45 sterlet larval head to illustrate the position and embryonic origin of cranial neuromast lines (shown as individual neuromasts) and ampullary organ fields (represented by coloured patches). The colour-coding indicates their lateral line placode (LLp) of origin, following *Gibbs and Northcutt, 2004*: blue, anterodorsal LLp (light blue, supraorbital; dark blue, infraorbital); orange, anteroventral LLp (preopercular); green, otic LLp (otic); pink, supratemporal LLp (supratemporal); light grey, middle LLp (middle). The black asterisk indicates the spiracular opening (first pharyngeal cleft). (**C–E**) Bar charts summarising different aspects of neuromast line disruption in *Foxg1* crispants, scored separately on left and right sides of the head for the supraorbital, infraorbital, otic, and preopercular neuromast lines. (The post-otic neuromast lines—supratemporal, middle, and posterior—were not scored.) Source data and a summary table are provided in *Supplementary file 3*. (**C**) Bar chart showing the percentage of all phenotypic *Foxg1* crispant sides (n=74) in which each neuromast line was disrupted. The otic line was the most commonly disrupted; the preopercular line was the least often disrupted. (**D**) Severity of phenotype. The bar chart shows the percentage of each disrupted neuromast line for which the phenotype was scored as severe (more than two-thirds of the line affected), moderate (between two-thirds and one-third of the line affected), or mild (less than one-third of the line affected). The preopercular line always had a mild phenotype; the other neuromast lines showed varying phenotypic severity. (**E**) Bar chart showing the percentage of each disrupted neuromast line with the following phenotypes: any ectopic ampullary organs (AOs); electroreceptors (ER) inside neuromasts (NMs); any neuromast (NM) gaps; ectopic ampullary organs plus neuromast gaps; ectopic ampullary organs alone; neuromast gaps alone. The otic and preopercular lines had almost opposite phenotypes: the otic line was always disrupted by gaps and only sometimes by ectopic ampullary organs as well (note the small size of the associated dorsal otic ampullary organ field in panel **A**), whereas the preopercular line was almost always disrupted by ectopic ampullary organs and only sometimes by gaps. The supraorbital and infraorbital lines were usually disrupted by ectopic ampullary organs (note the large size of the associated ampullary organ fields in panel **A**) but often also by gaps. Abbreviations for neuromast lines: io, infraorbital; m, middle; ol, otic; pop, preopercular; so, supraorbital; st, supratemporal. Abbreviations for anatomical landmarks: b, barbel; e, eye; m, mouth; n, naris; ot, otic vesicle; s, spiracle (first pharyngeal cleft). Scale bar: 200 µm.

ganglia; *Modrell et al., 2011b*.) Targeting multiple *Neurod* genes for CRISPR/Cas9-mediated mutagenesis in the future may shed light on the role played by this transcription factor family in ampullary organ development.

## FoxG1 represses electroreceptor formation in the neuromast-forming central zone of lateral line sensory ridges, whether directly or indirectly

We also targeted *Foxg1*, a mechanosensory-restricted transcription factor gene that we recently identified in the developing lateral line system of paddlefish and sterlet (*Minařík et al., 2024*). *Foxg1* is expressed in the central zones of lateral line sensory ridges where neuromasts form, though excluded from the central domains of neuromasts where hair cells differentiate (*Minařík et al., 2024*). Targeting *Foxg1* for CRISPR/Cas9-mediated mutagenesis in G0 sterlet embryos led to a striking phenotype: the formation within neuromast lines of ectopic electroreceptors, often in the large clusters normally seen in ampullary organs. In some cases, ampullary organ fields, which normally flank neuromast lines, effectively 'merged' across missing neuromast lines. This phenotype was revealed by examining expression of the electroreceptor-specific marker *Kcnab3*, and two ampullary organ-restricted transcription factor genes: *Mafa* (*Minařík et al., 2024*) and *Neurod4* (*Modrell et al., 2017a*). In a subset of these crispants, small clusters of electroreceptors were observed to form within existing neuromasts. Thus, FoxG1 seems to repress an ampullary organ fate within the central domain of lateral line sensory ridges where neuromasts form, and to repress electroreceptor formation within neuromasts.

Post-ISH Sox2 immunostaining of *Foxg1* crispants also revealed some missing sections of neuromast lines (particularly in the otic neuromast line), without ectopic electroreceptors or merging of ampullary organ fields. This suggests that *Foxg1* is also necessary for neuromast formation and/or maintenance. Indeed, after this manuscript was submitted, *foxg1a* expression was reported in the migrating posterior lateral line primordium and neuromasts of zebrafish (*Bell et al., 2024*). Loss of *foxg1a* function resulted in slower primordium migration, a temporary reduction in the number of neuromasts, reduced neuromast cell proliferation and fewer hair cells during both development and regeneration (*Bell et al., 2024*).

In the mouse inner ear, *Foxg1* is expressed in the prospective cochlea and all sensory patches, in hair cell progenitors and supporting cells (*Pauley et al., 2006*; *Tasdemir-Yilmaz et al., 2021*), plus a subset of hair cells (*Pauley et al., 2006*). Knockout leads to a shorter cochlea with extra rows of hair cells, and to loss or reduction of vestibular end organs (*Pauley et al., 2006*; *Hwang et al., 2009*). More hair cells and fewer supporting cells were seen after conditional knockdown of *Foxg1* in neonatal cochlear supporting cells, possibly via the transdifferentiation of supporting cells (*Zhang et al., 2019*; *Zhang et al., 2020*). This suggests the possibility that FoxG1 may act in the central zone of lateral line sensory ridges to maintain a proliferative progenitor state, as it does in the mouse olfactory epithelium (*Kawauchi et al., 2009*).

Furthermore, Fox family members can act as pioneer factors as well as transcription factors (*Golson and Kaestner, 2016*; *Lukoseviciute et al., 2018*). A pioneer factor role has been proposed for FoxI3 in otic placode development (see *Singh and Groves, 2016*). In the developing neural crest, FoxD3 acts early as a pioneer factor, opening enhancers and repositioning nucleosomes to prime genes controlling neural crest specification and migration and, concurrently, to repress the premature differentiation of, for example, melanocytes (*Lukoseviciute et al., 2018*). Later in neural crest development, FoxD3 represses enhancers associated with mesenchymal, neuronal and melanocyte lineages (*Lukoseviciute et al., 2018*). In cortical progenitors, FoxG1 suppresses the adoption at later stages of an early-born cell fate, namely, Cajal-Retzius cells (*Hanashima et al., 2004*). In the developing chicken otic placode, FoxG1 represses markers of other lineages, such as the olfactory and lens placodes, and epidermis (*Anwar et al., 2017*). Hence, it is possible that FoxG1 acts in the central zone of lateral line sensory ridges in electroreceptive fishes as a pioneer factor for neuromast/hair cell formation, and/or that it represses ampullary organ/electroreceptor formation.

The formation of ectopic ampullary organs/electroreceptors within neuromast lines in *Foxg1* crispants could be explained by various scenarios. FoxG1 might act as a direct repressor of electrosensory fate, its loss leading to ampullary organ formation instead of neuromasts (and to ectopic electoreceptor formation within neuromasts, as we observed in a few cases). Alternatively, given that ampullary organs normally form later than neuromasts, and neuromasts are missing in the absence of FoxG1 (see also *Bell et al., 2024*), the expansion of ampullary organ fields could be caused

indirectly by the loss of inhibitory signals from neuromasts that normally repress ampullary organ development. The fact that we often saw sections of missing neuromasts without ectopic ampullary organs might argue against this indirect role. However, the 'missing neuromasts only' phenotype was primarily seen in the otic neuromast line, where ampullary organ-promoting signals might be more restricted, as the associated ampullary organ field is small and only forms on the dorsal side of part of the line (see schematic in *Figure 9A*). It is also possible that FoxG1 acts via both of these hypothetical mechanisms. Consistent with this possibility, FoxG1 was recently reported to play a dual role in regulating neurogenesis versus gliogenesis in the cortex (*Bose et al., 2025*). In cortical progenitors, FoxG1 maintains neurogenesis and suppresses gliogenesis cell-autonomously by repressing expression of *Fgfr3*, encoding a receptor for the pro-gliogenic Fgf signaling pathway (*Bose et al., 2025*). In post-mitotic neurons, FoxG1 regulates (directly or indirectly) the expression of Fgf ligand genes, suggesting a non-cell-autonomous role for neuron-expressed FoxG1 in regulating gliogenesis in progenitors via Fgf signaling (*Bose et al., 2025*). Thus, it is possible that FoxG1 plays both cell-autonomous and non-cell-autonomous roles in regulating sensory organ and receptor cell formation in the central zone of lateral line sensory ridges.

Overall, these data lead us to propose a highly speculative hypothesis, namely that electrosensory organs may be the 'default' developmental fate within lateral line sensory ridges in electroreceptive vertebrates, and that FoxG1 represses this fate, whether directly or indirectly, to enable mechanosensory neuromasts and hair cells to form. (This speculation relates solely to developmental mechanisms: electroreceptors most likely evolved in the vertebrate ancestor via the diversification of lateral line hair cells; see *Baker, 2019*.) To test these hypotheses directly, it will be important in the future to identify global changes in gene expression and chromatin accessibility in the absence of FoxG1, as well as analyse sensory cell identity within individual lateral line organs using multiple molecular markers across a range of developmental stages.

## Summary and perspective

Overall, we have found that the 'hair cell' transcription factor Atoh1 is required for the formation of lateral line electroreceptors as well as hair cells, consistent with a close developmental relationship between these putative 'sister cell' types. Electrosensory-restricted Neurod4 may act redundantly with other Neurod family members expressed in developing ampullary organs. Mechanosensory-restricted FoxG1 represses the formation of electroreceptors within neuromast lines, whether directly or indirectly, suggesting the surprising possibility that electroreceptors may be the 'default' fate within lateral line sensory ridges.

## Materials and methods

### Animals

Fertilised sterlet (*A. ruthenus*) eggs were obtained from the breeding facility at the Research Institute of Fish Culture and Hydrobiology, Faculty of Fisheries and Protection of Waters, University of South Bohemia in České Budějovice, Vodňany, Czech Republic, and staged according to *Dettlaff et al., 1993*. For detailed information about sterlet husbandry, in vitro fertilisation and the rearing of embryos and yolk-sac larvae, see *Stundl et al., 2022*. Each fertilisation used a mix of sperm from three different males, so each batch was a mix of siblings and half-siblings. Upon reaching the desired developmental stages, embryos and yolk-sac larvae were euthanised by overdose of MS-222 (Sigma-Aldrich) and fixed in modified Carnoy's fixative (6 volumes 100% ethanol: 3 volumes 37% formaldehyde: 1 volume glacial acetic acid) for 3 hours at room temperature, dehydrated stepwise into 100% ethanol and stored at –20°C.

All experimental procedures were approved by the Animal Research Committee of the Faculty of Fisheries and Protection of Waters in Vodňany, University of South Bohemia in České Budějovice, Czech Republic, and by the Ministry of Agriculture of the Czech Republic (reference number: MSMT-12550/2016-3). Experimental fish were maintained according to the principles of the European Union (EU) Harmonized Animal Welfare Act of the Czech Republic, and Prizinciples of Laboratory Animal Care and National Laws 246/1992 'Animal Welfare' on the protection of animals.

## CRISPR guide RNA design and synthesis

Target gene sequences were identified using the National Center for Biotechnology Information (NCBI) Basic Local Alignment Search Tool BLAST (https://blast.ncbi.nlm.nih.gov/Blast.cgi; *McGinnis and Madden, 2004*) to search sterlet transcriptomic data (available at DDBJ/EMBL/GenBank under the accessions GKLU00000000 and GKEF01000000; see *Minařík et al., 2024*) or draft genomic sequence data (MH, unpublished) with the relevant paddlefish sequence (*Modrell et al., 2017a*). Transcriptomic sequence data were searched for *Tyr*, *Atoh1*, and *Neurod4*; genomic sequence data were searched for *Foxg1*. Chromosome-level sterlet genomes became available only after the project started: *Du et al., 2020* and the 2022 reference genome (GCF_902713425.1). Open-reading frames were identified using the NCBI ORF Finder tool (https://www.ncbi.nlm.nih.gov/orffinder/) and exons annotated by comparison with reference anamniote species (*Lepisosteus oculatus*, *Danio rerio*, *Xenopus tropicalis*) available via Ensembl (https://www.ensembl.org; *Cunningham et al., 2022*). Conserved domains were identified using NCBI BLASTX (https://blast.ncbi.nlm.nih.gov/Blast.cgi; *McGinnis and Madden, 2004*). Single-guide (sg) RNAs were preferentially designed to target 5' exons, ideally upstream of or within regions encoding known functional domains, to increase the probability of disrupting gene function. SgRNAs were designed using the CRISPR Guide RNA Design Tool from Benchling (https://benchling.com) and selected for synthesis based on the following criteria: (1) a high on-target score, ideally no less than 0.5; (2) no off-target matches identified within coding sequences in transcriptome and genome data, unless there were at least two mismatches in the 3' seed sequence (8–10 bp upstream of the protospacer adjacent motif [PAM], or in the PAM itself); and (3) if multiplexing, the sgRNA pair was ideally within 50–150 bases of each other to increase the probability of fragment deletion.

DNA templates for CRISPR sgRNAs were synthesised using plasmid pX335-U6-Chimeric_BB-CBh-hSpCas9n(D10A) (Addgene, plasmid #42335; *Cong et al., 2013*) containing the sgRNA scaffold. The sgRNA scaffold was amplified using a specific forward primer for each sgRNA, with an overhang containing the sgRNA target sequence and T7 promoter, and a reverse primer that was identical for all reactions (AAAAAAGCACCGACTCGGTGCC; personal communication, Dr Ahmed Elewa, Karolinska Institutet, Stockholm, Sweden). To enable direct synthesis of the sgRNAs from the PCR products, an overhang was added to the forward primer that contained the T7 promoter, followed by the 20-nucleotide sgRNA target sequence: GATCAC<u>TAATACGACTCACTATA</u>(20N)GTTTTAGAGCTA GAAAT, where the underlining indicates the T7 promoter and '(20N)' represents the specific sgRNA target sequence, followed by the plasmid-specific primer sequence (Addgene, plasmid #42335; *Cong et al., 2013*). If the first nucleotide of the sgRNA target sequence was G, this completed the T7 promoter (and became the first base of the sgRNA). For sgRNA target sequences that did not start with G, an additional G was added before the sgRNA target sequence to complete the T7 promoter, to ensure efficient transcription. The DNA template was amplified using Q5 polymerase (New England Biolabs, NEB) and purified using the Monarch PCR & DNA Cleanup Kit (NEB). The sgRNAs were synthesised using the HiScribe T7 High Yield RNA Synthesis Kit (NEB) and purified using the Monarch RNA Cleanup Kit (NEB) and stored at –80°C before use. Alternatively, chemically modified synthetic sgRNAs were ordered directly from Synthego (CRISPRevolution sgRNA EZ Kit).

## Embryo injection

On the day of injection, 1200 ng sgRNA were mixed with 2400 ng Cas9 protein with NLS (PNA Bio) in 5 µl nuclease-free water and incubated for 10 minutes at room temperature to form ribonucleoprotein (RNP) complexes. For sgRNA multiplexing, each sgRNA was pre-complexed with Cas9 protein in a separate tube, to prevent binding competition. Two RNP mixes were then combined 1:1 to a final volume of 5 µl, and 0.5 µl of 10% 10,000 MW rhodamine dextran (Invitrogen) added to better visualise the injection mixture and allow selection of properly injected embryos using rhodamine fluorescence. Injection mixtures were kept on ice throughout the injection session. 50 µl glass microcapillaries (Drummond Microcaps) were pulled in a capillary needle puller (PC-10, Narishige) set to 58°C with two light and one heavy weights, in single-stage pulling mode. Fertilised sterlet eggs were manually dechorionated using Dumont #5 forceps. A 1000 µl pipette tip cut to the same diameter as a dechorionated sterlet egg was used to prepare a series of wells in an agar plate to allow ideal egg positioning during injection using an automatic microinjector (FemtoJet 4x, Eppendorf), set to 100 hPa. Approximately 20 nl of the injection mixture (corresponding to approximately 4.8 ng sgRNAs and

9.6 ng Cas9) were injected into fertilised eggs or two-cell stage embryos, targeting the animal pole at a 45° angle. Injected embryos were moved to a clean Petri dish and, for optimum Cas9 efficiency, kept at room temperature until the end of the injecting session or until at least the 32-cell stage, then moved to a 16°C incubator. No more than 30 eggs were kept per 90 mm Petri dish. Unfertilised eggs and dead embryos were removed at the end of the injection day. Petri dishes were checked regularly for dead embryos and the water was changed at least twice a day before gastrulation was completed, and once daily post-gastrulation. Hatched larvae were kept for approximately 16 days post fertilisation until stage 45, then euthanised by MS-222 overdose and fixed with modified Carnoy's fixative (see above). Fixed larvae were then dehydrated stepwise into 100% ethanol and stored at –20°C.

## Gene cloning, in situ hybridisation, and immunohistochemistry

Total RNA was isolated from sterlet larvae using Trizol (Invitrogen, Thermo Fisher Scientific), following the manufacturer's protocol, and cDNA synthesised using the Superscript III First Strand Synthesis kit (Invitrogen, Thermo Fisher Scientific). We used our sterlet transcriptome assemblies (from pooled yolk-sac larvae at stages 40–45; *Minařík et al., 2024*), which are available at DDBJ/EMBL/GenBank under the accessions GKLU00000000 and GKEF01000000, to design primers for *Neurod4* (forward: GAGAGAGCCCCAAAGAGACGAG; reverse: CTGCTTGAGCGAGAAGTTGACG). cDNA fragments amplified under standard PCR conditions were cloned into the pDrive cloning vector (QIAGEN). Individual clones were verified by sequencing (Department of Biochemistry Sequencing Facility, University of Cambridge, UK) and sequence identity verified using NCBI BLAST (https://blast.ncbi.nlm.nih.gov/Blast.cgi; *McGinnis and Madden, 2004*). For *Neurod1*, *Neurod2*, *Neurod4*, and *Neurod6*, synthetic gene fragments based on sterlet transcriptome data, with added M13 forward and reverse primer adaptors, were ordered from Twist Bioscience. GenBank accession numbers are as follows: *Neurod1* OQ808944, *Neurod2* OQ808945, *Neurod4* OQ808946, *Neurod6* OQ808947. The other genes used in this study have been published (*Minařík et al., 2024*).

The sterlet riboprobe template sequences were designed before chromosome-level genome assemblies for sterlet were available (*Du et al., 2020* and the 2022 reference genome: GCF_902713425.1). Genome analysis showed that an independent whole-genome duplication occurred in the sterlet lineage, from which approximately 70% of ohnologs (i.e., gene paralogs arising from the whole-genome duplication) have been retained (*Du et al., 2020*). *Supplementary file 4* shows the percentage match for each *Neurod* family riboprobe with the two ohnologs, obtained by performing a nucleotide BLAST search against the reference genome (GCF_902713425.1). The percentage match with the 'targeted' *Neurod* family ohnolog ranged from 98.9% to 100%; the percentage match with the second ohnolog was also high, ranging from 96.2% to 100% (*Supplementary file 1*), suggesting that transcripts from the second ohnolog are likely to be targeted by each of these riboprobes. *Supplementary file 4* shows the equivalent data for a previously published paddlefish *Cacna1d* riboprobe (*Modrell et al., 2017a*) used for two *Atoh1* crispants only. Equivalent data for the other riboprobes used in this study are available in *Minařík et al., 2024*.

Digoxigenin-labelled riboprobes were synthesised as previously described (*Minařík et al., 2024*). Whole-mount ISH was performed as previously described (*Modrell et al., 2011a*). For a few crispants, fluorescein-labelled riboprobes were synthesised using Fluorescein RNA Labelling Mix (Roche) and the ISH was performed using anti-fluorescein-AP Fab fragments (Roche) and SIGMAFAST Fast Red tablets (Sigma). Whole-mount immunostaining post-ISH for Sox2 (rabbit monoclonal antibody, 1:200; ab92494, Abcam) was performed as previously described (*Metscher and Müller, 2011*), using a horseradish peroxidase-conjugated goat anti-rabbit antibody (1:300, Jackson ImmunoResearch) and the metallographic peroxidase substrate EnzMet kit (Nanoprobes) as per the manufacturer's instructions.

## Genotyping

To confirm successful mutation in targeted regions, genotyping was performed on trunk and tail tissue that had been removed before ISH and stored in 100% ethanol at –20°C. The tissue was digested using Rapid Extract Lysis Kit (PCR Biosystems) and the target region was amplified using HS Taq Mix Red (PCR Biosystems) according to the manufacturer's instructions. Primers used for genotyping are listed in *Supplementary file 1*. After agarose gel electrophoresis, PCR products were extracted using the MinElute Gel Extraction Kit (QIAGEN) and submitted for sequencing (Genewiz by Azenta Life

Sciences). To analyse CRISPR editing efficiency, Sanger trace files were uploaded to the Synthego Inference of CRISPR Edits (ICE) tool (https://ice.synthego.com; *Conant et al., 2022*).

### Imaging and image processing

Larvae were placed in a slit in an agar-coated Petri dish with PBS and imaged using a Leica MZFLIII dissecting microscope equipped with either a MicroPublisher 5.0 RTV camera (QImaging) controlled by QCapture Pro 6.0 or 7.0 software (QImaging), or a MicroPublisher 6 colour CCD camera (Teledyne Photometrics) controlled by Ocular software (Teledyne Photometrics). For most larvae, a stack of images was acquired by manually focusing through the sample, then Helicon Focus software (Helicon Soft Limited) was used for focus stacking. Fragments of skin for skin-mount imaging were obtained by embedding larval heads in an oil-based modeling clay (Koh-I-Noor Hardtmuth) on a Petri dish and dissecting them under PBS using an ophthalmic scalpel (P-715, FEATHER Safety Razor Co. Ltd). Skin fragments were slide-mounted and coverslipped using Fluoroshield mounting medium with DAPI (Sigma-Aldrich). They were imaged using a Zeiss AxioSkop 2 microscope equipped with a MicroPublisher 6 colour CCD camera (Teledyne Photometrics) controlled by Ocular software (Teledyne Photometrics). Adobe Photoshop (Adobe Systems Inc) was used to process images.

### Acknowledgements

Thanks to Melinda Modrell for pilot work on CRISPR/Cas9 in paddlefish and for her advice and helpful discussions in planning the project. Thanks to Marek Rodina and Martin Kahanec for their help with sterlet spawns. Thanks to David Jandzik, András Simon, Alberto Joven Araus, Ahmed Elewa, and Mustafa Khokha for invaluable advice on CRISPR/Cas9 approaches and genotyping. Thanks to Jan Stundl and David Jandzik for sharing their sterlet *Tyr* sgRNA sequences prior to publication. Thanks to Christine Hirschberger and Rolf Ericsson for their help with some of the in situ hybridisation rounds. This work was supported by the Biotechnology and Biological Sciences Research Council (BBSRC: grant BB/P001947/1 to CB). Additional support for MM was provided by the Cambridge Isaac Newton Trust (grant 20.07[c] to CB) and by the School of the Biological Sciences, University of Cambridge. AC was supported by a PhD research studentship from the Anatomical Society with additional funding from the Cambridge Philosophical Society. The work of RF, MV and MP was supported by the Ministry of Education, Youth and Sports of the Czech Republic projects CENAKVA (LM2018099) and Biodiversity (CZ.02.1.01/0.0/0.0/16_025/0007370) and by the Czech Science Foundation (22–31141 J).

## Additional information

### Funding

| Funder | Grant reference number | Author |
|---|---|---|
| Biotechnology and Biological Sciences Research Council | BB/P001947/1 | Clare VH Baker |
| Isaac Newton Trust | Grant 20.07 (c) | Clare VH Baker |
| Anatomical Society | Research Studentship | Alexander S Campbell Clare VH Baker |
| Cambridge Philosophical Society | Research Studentship | Alexander S Campbell |
| School of the Biological Sciences, University of Cambridge | | Martin Minařík |
| Ministry of Education, Youth and Sports of the Czech Republic | CENAKVA LM2018099 | Roman Franěk Michaela Vazačová Martin Pšenička |

| Funder | Grant reference number | Author |
|---|---|---|
| Ministry of Education, Youth and Sports of the Czech Republic | Biodiversity CZ.02.1.01/0.0/0.0/16_025/0007370 | Roman Franěk Michaela Vazačová Martin Pšenička |
| Czech Science Foundation | Project 22-31141J | Roman Franěk Michaela Vazačová Martin Pšenička |

The funders had no role in study design, data collection and interpretation, or the decision to submit the work for publication.

## Author contributions

Martin Minařík, Conceptualization, Investigation, Visualization, Methodology, Writing – original draft, Project administration, Writing – review and editing; Alexander S Campbell, Roman Franěk, Michaela Vazačová, Investigation, Writing – review and editing; Miloš Havelka, David Gela, Resources, Writing – review and editing; Martin Pšenička, Resources, Funding acquisition, Writing – review and editing; Clare VH Baker, Conceptualization, Supervision, Funding acquisition, Writing – original draft, Project administration, Writing – review and editing

## Author ORCIDs

Martin Minařík ![ORCID] https://orcid.org/0000-0001-6660-0031
Alexander S Campbell ![ORCID] https://orcid.org/0009-0003-1539-214X
Roman Franěk ![ORCID] https://orcid.org/0000-0002-3464-1872
Miloš Havelka ![ORCID] http://orcid.org/0000-0003-3574-6243
Martin Pšenička ![ORCID] http://orcid.org/0000-0002-3808-7856
Clare VH Baker ![ORCID] https://orcid.org/0000-0002-4434-3107

## Ethics

Sterlet animal work was reviewed and approved by The Animal Research Committee of Research Institute of Fish Culture and Hydrobiology, Faculty of Fisheries and Protection of Waters, University of South Bohemia in České Budějovice, Vodňany, Czech Republic and Ministry of Agriculture of the Czech Republic (MSMT-12550/2016–3).

## Decision letter and Author response

Decision letter https://doi.org/10.7554/eLife.96285.sa1
Author response https://doi.org/10.7554/eLife.96285.sa2

## Additional files

### Supplementary files

Supplementary file 1. Breakdown by sgRNA mix of CRISPR/Cas9 experiments. For each sgRNA mix targeting *tyrosinase*, *Atoh1*, *Neurod4*, or *Foxg1*, the table shows the number of independent batches, markers used for analysis, percentage with phenotypes, and genotyping information including the genotyping primer sequences. The table also shows total numbers and percentage scores across each of the targeted genes. Most of the *Tyr* crispant data are shared between this study and *Campbell et al., 2025*. *Tyr* sgRNAs 7 and 8 were designed and published by *Stundl et al., 2022* as their *tyr* sgRNA 3 and *tyr* sgRNA 4, respectively.

Supplementary file 2. Breakdown by individual crispant of *Atoh1* crispant phenotypes. For each of 64 *Atoh1* crispants, the table shows the batch number, riboprobe used, and the laterality and severity of any phenotype as scored in lateral view on left and right sides independently, where 'severe' means more than two-thirds of cranial hair cells/electroreceptors were absent; 'moderate' means between one-third and two-thirds of cranial hair cells/electroreceptors were absent; 'mild' means less than one-third of cranial hair cells/electroreceptors were absent. The table also shows total numbers and percentage scores across all crispants. Crispants are ordered by riboprobe used, and within this by the presence or absence of a phenotype (with non-phenotypic crispants in blue font).

Supplementary file 3. Breakdown by individual crispant of *Foxg1* crispant phenotypes. For each of 87 *Foxg1* crispants, the table shows the sgRNA mix used, batch number, riboprobe used, type of

phenotype, if any (neuromast lines disrupted by ectopic ampullary organs, electroreceptors inside neuromasts, and/or neuromast gaps), and the laterality and severity of the phenotype as scored in lateral and ventral view for each of the pre-otic neuromast lines (supraorbital, infraorbital, otic and preopercular neuromast lines) on the left and right sides of the head. The phenotype was classed as 'severe' when more than two-thirds of the neuromast line was affected; 'moderate' when between one-third and two-thirds of the line was affected; and 'mild' when less than one-third of the line was affected. Crispants are ordered by sgRNA mix, then by riboprobe used, and within this by the presence or absence of a phenotype (with non-phenotypic crispants in blue font). The table also shows summary percentage data across crispants and individual neuromast lines.

Supplementary file 4. Riboprobe information. For each riboprobe used for in situ hybridisation for *Neurod* family genes, the table shows the primer sequences used for cloning the cDNA template, the GenBank accession number of the top-matched ohnolog and the nucleotide range targeted by the riboprobe. The table also shows the chromosomal location, riboprobe percentage identity and genome annotation of both the top-matched ohnolog and the second ohnolog. Equivalent information is given for a previously published paddlefish (*Polyodon spathula*) *Cacna1d* riboprobe (*Modrell et al., 2017a*) used to analyse two *Atoh1* sterlet crispants.

MDAR checklist

## Data availability

The original data required to reproduce the claims of the paper are provided in the manuscript and supplementary figures, together with additional image files deposited into the Dryad database (https://doi.org/10.5061/dryad.fqz612k3s). Previously published sterlet transcriptome assemblies (from pooled yolk-sac larvae at stages 40-45; *Minařík et al., 2024*) are available at DDBJ/EMBL/GenBank under the accessions GKLU00000000 and GKEF01000000. Previously published paddlefish RNA-seq data (from pooled paddlefish opercula and fin tissue at stage 46; *Modrell et al., 2017a*) are available via the NCBI Gene Expression Omnibus (GEO) database (https://www.ncbi.nlm.nih.gov/geo/) under accession code GSE92470.

The following dataset was generated:

| Author(s) | Year | Dataset title | Dataset URL | Database and Identifier |
|---|---|---|---|---|
| Minařík M, Campbell AS, Franěk R, Vazačová M, Havelka M, Gela D, Pšenička M, Baker CVH | 2025 | Data for: Atoh1 is required for the formation of lateral line electroreceptors and hair cells, whereas Foxg1 represses an electrosensory fate | https://doi.org/10.5061/dryad.fqz612k3s | Dryad Digital Repository, 10.5061/dryad.fqz612k3s |

The following previously published datasets were used:

| Author(s) | Year | Dataset title | Dataset URL | Database and Identifier |
|---|---|---|---|---|
| Minařík M, Modrell MS, Gillis JA, Campbell AS, Fuller I, Lyne R, Micklem G, Gela D, Pšenička M, Baker CVH | 2023 | TSA: Acipenser ruthenus, transcriptome shotgun assembly | https://www.ncbi.nlm.nih.gov/nuccore/GKLU00000000.1 | NCBI GenBank, GKLU00000000.1 |
| Minařík M, Modrell MS, Gillis JA, Campbell AS, Fuller I, Lyne R, Micklem G, Gela D, Pšenička M, Baker CVH | 2023 | TSA: Acipenser ruthenus, transcriptome shotgun assembly | https://www.ncbi.nlm.nih.gov/nuccore/GKEF00000000.1 | NCBI GenBank, GKEF00000000.1 |
| Modrell MS, Lyne M, Carr AR, Zakon HH, Buckley D, Campbell AS, Davis MC, Micklem G, Baker CVH | 2017 | Insights into electrosensory organ development, physiology and evolution from a lateral line organ-enriched transcriptome | http://www.ncbi.nlm.nih.gov/geo/query/acc.cgi?acc=GSE92470 | NCBI Gene Expression Omnibus, GSE92470 |

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

# Appendix 1

## Appendix 1—key resources table

| Reagent type (species) or resource | Designation | Source or reference | Identifiers | Additional information |
|---|---|---|---|---|
| Gene (*Acipenser ruthenus*) | *Atoh1* | NCBI_Gene (RRID:SCR_002473) | GeneID:117420670 | |
| Gene (*A. ruthenus*) | *Cacna1d* | NCBI_Gene (RRID:SCR_002473) | GeneID:117413950 | |
| Gene (*A. ruthenus*) | *Foxg1* | NCBI_Gene (RRID:SCR_002473) | GeneID:117418845 | |
| Gene (*A. ruthenus*) | *Gfi1* | NCBI_Gene (RRID:SCR_002473) | GeneID:117408280 | |
| Gene (*A. ruthenus*) | *Kcnab3* | NCBI_Gene (RRID:SCR_002473) | GeneID:117404443 | |
| Gene (*A. ruthenus*) | *Mafa* | NCBI_Gene (RRID:SCR_002473) | GeneID:117399627 | |
| Gene (*A. ruthenus*) | *Neurod1* | NCBI_Gene (RRID:SCR_002473) | GeneID:117426329 | |
| Gene (*A. ruthenus*) | *Neurod2* | NCBI_Gene (RRID:SCR_002473) | GeneID:117433279 | |
| Gene (*A. ruthenus*) | *Neurod4* | NCBI_Gene (RRID:SCR_002473) | GeneID:131720860 | |
| Gene (*A. ruthenus*) | *Neurod6* | NCBI_Gene (RRID:SCR_002473) | GeneID:117435768 | |
| Gene (*A. ruthenus*) | *Pou4f3* | NCBI_Gene (RRID:SCR_002473) | GeneID:117968545 | |
| Biological sample (*A. ruthenus*) | Fertilised sterlet sturgeon eggs and embryos/yolk-sac larvae (*A. ruthenus*) | Research Institute of Fish Culture and Hydrobiology, Faculty of Fisheries and Protection of Waters, University of South Bohemia in České Budějovice Vodňany, Czech Republic | | |
| Antibody | Sox2 antibody (rabbit monoclonal) | Abcam | Cat.#:ab92494; RRID:AB_10585428 | (1:200) |
| Antibody | horseradish peroxidase-conjugated goat anti-rabbit IgG (H+L) | Jackson ImmunoResearch | Cat.#:111-035-003; RRID:AB_2313567 | (1:300) |
| Antibody | sheep anti-digoxigenin Fab fragments, AP-conjugated | Roche | Cat.#:11093274910; RRID:AB_514497 | (1:2000) |
| Antibody | sheep anti-fluorescein Fab fragments, fluorescein conjugated | Roche | Cat.#:11426338910; RRID:AB_514504 | (1:1000) |
| Recombinant DNA reagent | pX335-U6-Chimeric_BB-CBh-hSpCas9n(D10A) (plasmid) | Addgene (*Cong et al., 2013*) | RRID:Addgene_42335 | Used to synthesize DNA templates containing the single guide (sg)RNA scaffold |
| Sequence-based reagent | *Atoh1* riboprobe forward primer (F) | *Minařík et al., 2024* | PCR primers | AGCTCGCAGGAGGAGATGCACA |
| Sequence-based reagent | *Atoh1* riboprobe reverse primer (R) | *Minařík et al., 2024* | PCR primers | TGGTGTGGTTCTGGAGTTTGAGT |
| Sequence-based reagent | *Cacna1d* riboprobe F | *Minařík et al., 2024* | PCR primers | TACCAGGAGCTCATGTGCAG |
| Sequence-based reagent | *Cacna1d* riboprobe R | *Minařík et al., 2024* | PCR primers | CAATGCCAACCTCAACAATG |
| Sequence-based reagent | *Foxg1* riboprobe F | *Minařík et al., 2024* | PCR primers | TCAGCTCCTGAGGTCCAACT |
| Sequence-based reagent | *Foxg1* riboprobe R | *Minařík et al., 2024* | PCR primers | CAGGCTCAGGTTGTGTCTGA |
| Sequence-based reagent | *Gfi1* riboprobe F | *Minařík et al., 2024* | PCR primers | TGAGACGGCTGACTTCTCCT |
| Sequence-based reagent | *Gfi1* riboprobe R | *Minařík et al., 2024* | PCR primers | GGCTGTGTGTGATCAGGTTG |
| Sequence-based reagent | *Kcnab3* riboprobe F | *Minařík et al., 2024* | PCR primers | GGTAAATTCAGCGTGGAGGA |
| Sequence-based reagent | *Kcnab3* riboprobe R | *Minařík et al., 2024* | PCR primers | ACCTTCGATGATGTGCTTCC |
| Sequence-based reagent | *Neurod4* riboprobe F | This paper | PCR primers | GAGAGAGCCCCAAAGAGACGAG |
| Sequence-based reagent | *Neurod4* riboprobe R | This paper | PCR primers | CTGCTTGAGCGAGAAGTTGACG |

*Appendix 1 Continued on next page*

*Appendix 1 Continued*

| Reagent type (species) or resource | Designation | Source or reference | Identifiers | Additional information |
|---|---|---|---|---|
| Sequence-based reagent | *Pou4f3* riboprobe F | **Minařík et al., 2024** | PCR primers | GAGTTTGCCTTCCAAATCCA |
| Sequence-based reagent | *Pou4f3* riboprobe R | **Minařík et al., 2024** | PCR primers | TTGTTGTGGGACAAGGTCAA |
| Sequence-based reagent | M13 F | https://www.genewiz.com/en-GB/Public/Resources/Free-Universal-Primers | PCR primers | GTAAAACGACGGCCAG |
| Sequence-based reagent | M13 R with SP6 promoter sequence | https://www.genewiz.com/en-GB/Public/Resources/Free-Universal-Primers | PCR primers | ATTTAGGTGACACTATAG CAGGAAACAGCTATGAC |
| Sequence-based reagent | *Mafa* synthetic gene fragment | **Minařík et al., 2024** | GenBank:OR327047 | Ordered from Twist Biosciences with PCR primer adaptors attached (M13 F and M13 R with SP6 promoter sequence). See Materials and Methods. |
| Sequence-based reagent | *Neurod1* synthetic gene fragment | This paper | GenBank:OQ808944 | Ordered from Twist Biosciences with PCR primer adaptors attached (M13 F and M13 R with SP6 promoter sequence). See Materials and Methods. |
| Sequence-based reagent | *Neurod2* synthetic gene fragment | This paper | GenBank:OQ808945 | Ordered from Twist Biosciences with PCR primer adaptors attached (M13 F and M13 R with SP6 promoter sequence). See Materials and Methods. |
| Sequence-based reagent | *Neurod4* synthetic gene fragment | This paper | GenBank:OQ808946 | Ordered from Twist Biosciences with PCR primer adaptors attached (M13 F and M13 R with SP6 promoter sequence). See Materials and Methods. |
| Sequence-based reagent | *Neurod6* synthetic gene fragment | This paper | GenBank:OQ808947 | Ordered from Twist Biosciences with PCR primer adaptors attached (M13 F and M13 R with SP6 promoter sequence). See Materials and Methods. |
| Sequence-based reagent | sgRNA scaffold R | Pers. comm., Dr Ahmed Elewa, Karolinska Institutet, Stockholm, Sweden | PCR primers | AAAAAAGCACCGACTCGGTGCC |
| Sequence-based reagent | *Atoh1* sgRNA F1 | This paper | PCR primers | GATCACTAATACGACTCACTATAGACCTTGT AAAAGATCGGAAGTTTTAGAGCTAGAAAT |
| Sequence-based reagent | *Atoh1* sgRNA F2 | This paper | PCR primers | GATCACTAATACGACTCACTATAGCTTGTCAT TGTCAAATGACGTTTTAGAGCTAGAAAT |
| Sequence-based reagent | *Foxg1* sgRNA F1 | This paper | PCR primers | GATCACTAATACGACTCACTATAGAAACATCT TTTGCCCAACCGTTTTAGAGCTAGAAAT |
| Sequence-based reagent | *Foxg1* sgRNA F2 | This paper | PCR primers | GATCACTAATACGACTCACTATATCTTCCGA GCAAGGTAACTCGTTTTAGAGCTAGAAAT |
| Sequence-based reagent | *Foxg1* sgRNA F3 | This paper | PCR primers | GATCACTAATACGACTCACTATATGATGCT GAAGGACGACTTGGTTTTAGAGCTAGAAAT |
| Sequence-based reagent | *Foxg1* sgRNA F4 | This paper | PCR primers | GATCACTAATACGACTCACTATACTGGCTCG TCCTCGGGCCGGGTTTTAGAGCTAGAAAT |
| Sequence-based reagent | *Neurod4* sgRNA F1 | This paper | PCR primers | GATCACTAATACGACTCACTATAGGAGCGTT TCAAGGCCAGGCGTTTTAGAGCTAGAAAT |
| Sequence-based reagent | *Neurod4* sgRNA F2 | This paper | PCR primers | GATCACTAATACGACTCACTATAGTGAGCGT TCTCGCATGCACGTTTTAGAGCTAGAAAT |
| Sequence-based reagent | *Neurod4* sgRNA F3 | This paper | PCR primers | GATCACTAATACGACTCACTATAGCCTGG CCCACAACTACATCGTTTTAGAGCTAGAAAT |
| Sequence-based reagent | *Neurod4* sgRNA F4 | This paper | PCR primers | GATCACTAATACGACTCACTATAGAGGGGCC CCGAGAAGCTGCGTTTTAGAGCTAGAAAT |
| Sequence-based reagent | *Neurod4* sgRNA F5 | This paper | PCR primers | GATCACTAATACGACTCACTATAGTCTCCCC AGCCCTCCCTACGTTTTAGAGCTAGAAAT |
| Sequence-based reagent | *Neurod4* sgRNA F6 | This paper | PCR primers | GATCACTAATACGACTCACTATAGACAACCA CTCCCCGGATTGGTTTTAGAGCTAGAAAT |
| Sequence-based reagent | *Neurod4* sgRNA F7 | This paper | PCR primers | GATCACTAATACGACTCACTATAGACCCTGC GCAGGCTCTCCAGTTTTAGAGCTAGAAAT |

*Appendix 1 Continued on next page*

*Appendix 1 Continued*

| Reagent type (species) or resource | Designation | Source or reference | Identifiers | Additional information |
|---|---|---|---|---|
| Sequence-based reagent | *Neurod4* sgRNA F8 | This paper | PCR primers | GATCACTAATACGACTCACTATAGCAGCTGGGTCCCCTGCTGAGTTTTAGAGCTAGAAAT |
| Sequence-based reagent | *Neurod4* sgRNA F9 | This paper | PCR primers | GATCACTAATACGACTCACTATAGGGGCCGTGTGCTCAGGGATGTTTTAGAGCTAGAAAT |
| Sequence-based reagent | *Tyr* sgRNA F1 | This paper | PCR primers | GATCACTAATACGACTCACTATAGGTGCCAAGGCAAAAACGCTGTTTTAGAGCTAGAAAT |
| Sequence-based reagent | *Tyr* sgRNA F2 | This paper | PCR primers | GATCACTAATACGACTCACTATAGATATCCCTCCATACATTATGTTTTAGAGCTAGAAAT |
| Sequence-based reagent | *Tyr* sgRNA F3 | This paper | PCR primers | GATCACTAATACGACTCACTATAGATGTTTCTAAACATTGGGGGTTTTAGAGCTAGAAAT |
| Sequence-based reagent | *Tyr* sgRNA F4 | This paper | PCR primers | GATCACTAATACGACTCACTATAGCTATGAATTTATTTTTTTCGTTTTAGAGCTAGAAAT |
| Sequence-based reagent | *Tyr* sgRNA F5 | This paper | PCR primers | GATCACTAATACGACTCACTATAGCAAGGTATACGAAAGTTGAGTTTTAGAGCTAGAAAT |
| Sequence-based reagent | *Tyr* sgRNA F6 | This paper | PCR primers | GATCACTAATACGACTCACTATAGATTGCAAGTTCGGCTTCTTGTTTTAGAGCTAGAAAT |
| Sequence-based reagent | *Tyr* sgRNA F7 | **Stundl et al., 2022** | PCR primers | GATCACTAATACGACTCACTATAGGTTAGAGACTTTATGTAACGTTTTAGAGCTAGAAAT |
| Sequence-based reagent | *Tyr* sgRNA F8 | **Stundl et al., 2022** | PCR primers | GATCACTAATACGACTCACTATAGGCTCCATGTCTCAAGTCCAGTTTTAGAGCTAGAAAT |
| Sequence-based reagent | *Atoh1* genotyping F | This paper | PCR primers | GACACAGACAACAGCGAGGA |
| Sequence-based reagent | *Atoh1* genotyping R | This paper | PCR primers | ACGGGAACTGCGTTGTATTC |
| Sequence-based reagent | *Atoh1* genotyping R1/R2 | This paper | PCR primers | CTGCTTGAGCGAGAAGTTGACG |
| Sequence-based reagent | *Foxg1* genotyping F | This paper | PCR primers | GATTGAGGTCCAACTGTGCTGC |
| Sequence-based reagent | *Foxg1* genotyping R | This paper | PCR primers | GTCTGATGGAGTTTTGCCAGCC |
| Sequence-based reagent | *Neurod4* genotyping F1 | This paper | PCR primers | ATGTACGAGGAGGAGGAAGAGGAAG |
| Sequence-based reagent | *Neurod4* genotyping F2 | This paper | PCR primers | GAGAGAGCCCCAAAGAGACGAG |
| Sequence-based reagent | *Neurod4* genotyping R3 | This paper | PCR primers | AAGACCCAGAAGCTGTCCAA |
| Sequence-based reagent | *Neurod4* genotyping F4 | This paper | PCR primers | CTCCTGCTTGAGCGAGAAGT |
| Sequence-based reagent | *Tyr* genotyping F | This paper | PCR primers | GCGTCTCTCCAGTCCCAATA |
| Sequence-based reagent | *Tyr* genotyping R | This paper | PCR primers | AGAGAGAAGTGGCCCTTGGT |
| Peptide, recombinant protein | Cas9 protein with NLS | PNA Bio | Cat.#:CP01-200 | |
| Peptide, recombinant protein | Q5 High-Fidelity DNA Polymerase | New England Biolabs | Cat.#:M0491S | |
| Commercial assay or kit | Superscript III First Strand Synthesis kit | Invitrogen | Cat.#:18080051 | |
| Commercial assay or kit | Qiagen PCR cloning kit | Qiagen | Cat.#:231124 | |
| Commercial assay or kit | MinElute Gel Extraction Kit | Qiagen | Cat.#:28604 | |
| Commercial assay or kit | HS Taq Mix Red | PCR Biosystems | Cat.#:PB10.23–02 | |
| Commercial assay or kit | Monarch PCR & DNA Cleanup Kit | New England Biolabs | Cat.#:T1030 | |
| Commercial assay or kit | HiScribe T7 High Yield RNA Synthesis Kit | New England Biolabs | Cat.#:E2040S | |
| Commercial assay or kit | Monarch RNA Cleanup Kit | New England Biolabs | Cat.#:T2040 | |
| Commercial assay or kit | EnzMet kit | Nanoprobes | Cat.#:6010 | |
| Commercial assay or kit | Rapid Extract Lysis Kit | PCR Biosystems | Cat.#:PB15.11–08 | |

*Appendix 1 Continued*

| Reagent type (species) or resource | Designation | Source or reference | Identifiers | Additional information |
|---|---|---|---|---|
| Commercial assay or kit | Digoxigenin RNA Labelling Mix | Roche | Cat.#:11277073910 | |
| Commercial assay or kit | Fluorescein RNA Labelling Mix | Roche | Cat.#:11685619910 | |
| chemical compound, drug | NBT/BCIP | Roche | Cat.#:11681451001 | (1:50) |
| Chemical compound, drug | SIGMAFAST Fast Red tablets | Sigma | Cat.#:F4648-50SET | |
| Software, algorithm | NCBI BLAST | National Institutes of Health (NIH) | RRID:SCR_004870 | |
| Software, algorithm | NCBI ORF Finder | National Institutes of Health (NIH) | RRID:SCR_016643 | |
| Software, algorithm | Benchling | Benchling | RRID:SCR_013955 | |
| Software, algorithm | CRISPR Guide RNA Design Tool | Benchling | RRID:SCR_013955 | |
| Software, algorithm | Inference of CRISPR Edits (ICE) | Synthego | RRID:SCR_024508 | |
| Software, algorithm | Ocular | Teledyne Photometrics | RRID:SCR_024490 | |
| Software, algorithm | QCapture Pro 6.0 | QImaging | RRID:SCR_014432 | |
| Software, algorithm | QCapture Pro 7.0 | QImaging | RRID:SCR_014432 | |
| Software, algorithm | Helicon Focus | Helicon Soft | RRID:SCR_014462 | |
| Software, algorithm | Adobe Photoshop | Adobe | RRID:SCR_014199 | |
| Other | Ophthalmic scalpel | FEATHER Safety Razor Co. Ltd. | Cat.#:P-715 | |

