## [Editor Report]

Many aquatic vertebrates have a lateral line system comprising both mechanosensory and electrosensory receptor cells. Using a gene knockout approach in the sterlet, this study convincingly demonstrates that the development of both types of receptor cells depends on the transcription factor Atoh1, it further shows that another transcription factor, FoxG1, promotes mechanoreceptor but represses electroreceptor development. Particularly interesting from the evolutionary perspective is the resulting hypothesis of the authors that electrosensory organs may be the 'default' developmental fate in electroreceptive vertebrates. Overall, the study provides fundamental new insights into sensory development and evolution.

---

## [Decision Letter]

**Decision letter after peer review:**

Thank you for submitting your article "Atoh1 is required for the formation of lateral line electroreceptors and hair cells, whereas Foxg1 represses an electrosensory fate" for consideration by *eLife*. Your article has been reviewed by 3 peer reviewers, and the evaluation has been overseen by a Reviewing Editor and Kathryn Cheah as the Senior Editor.

Essential Revisions:

The reviewers agree that using Cas9/Crispr technology on the sterlet is a significant advancement of this unconventional model system on the highly interesting question of the development and evolution of mechano-/electroreceptors.

While the Atoh1 manipulation is overall well documented and convincing, the reviewers think that the following experiments are critical for a more solid manuscript:

1) Quantification and statistical assessment of results

2) The understanding of the phenotypes could be enhanced with better visual guidance, i.e. adding illustrative cartoons that depict the sensory structures and more "zoom-ins".

3) The authors need to consider alternative explanations for their phenotypes more seriously. Specifically, at least three different scenarios could explain the current findings:

a) a transformation of fate from neuromasts to ampullary organs due to the loss of FoxG1 as a repressor of ampullary fate (the authors' hypothesis); or (2) an expansion of ampullary precursors because of apoptosis of neuromasts; or (3) an expansion of ampullary precursors because of the absence of neuromasts that exert a repressive effect on ampullary organ development.

A better documentation of the Foxg1 effects on patterning, as well as an extended discussion of the different possible scenarios will be helpful. Please see the individual reviews of reviewers 2 and 3 for details on this.

*Reviewer #1 (Recommendations for the authors):*

In this study, the authors used CRISPR/Cas9-mediated mutagenesis in the sterlet, a small sturgeon species, to investigate the molecular mechanisms underlying the development of electroreceptors. The authors have provided high-quality data, and the paper is well-written with clarity.

The authors' key findings are:

Atoh1, a well-known 'hair cell' transcription factor, is required for the differentiation of both hair cells and electroreceptors, suggesting deep conservation of molecular mechanisms between these cell types.

The mechanosensory-restricted transcription factor Foxg1 appears to repress the formation of electroreceptors within neuromast lines. This leads the authors to propose the intriguing hypothesis that electrosensory organs may represent the 'default' fate within lateral line sensory ridges in electroreceptive vertebrates. It challenges the assumption that mechanosensory hair cells evolutionarily preceded electroreceptors. If further evidence supports this hypothesis, it could reshape our understanding of the evolutionary origins of these sensory cell types.

A notable limitation is the mosaic nature of the observed phenotypes in many experiments. This mosaicism complicates the interpretation of phenotypic effects and may underestimate the consequences of gene disruption. The authors acknowledge and discuss this limitation transparently.

In addition, this work represents a significant technical advance. The authors' use of CRISPR/Cas9 mutagenesis in a non-standard animal model electroreceptive species demonstrates the utility of this approach for functional studies in understudied systems.

I have a few suggestions to consider:

It would be beneficial to include quantitative data of the observed phenotypes. Identifying subtle differences between images can be challenging.

The current images could be enhanced with visual aids. I recommend adding illustrative cartoons that depict the sensory structures, like Campbell et al. 2024 bioRxiv publication. This would greatly aid in visual comprehension.

Distinguishing between neuromasts and ampullary organs, particularly the electroreceptor formation within neuromast lines, could be difficult. Close-up images or 'zoom-ins' in areas where differentiation is crucial would be extremely helpful.

*Reviewer #2 (Recommendations for the authors):*

This study uses CRISPR/Cas9-based gene editing to knockout genes encoding three transcription factors with potential roles in lateral line development in F_0_ embryos of the sterlet (Acipenser ruthenus). In this bony fish, related to sturgeons, the lateral line system comprises both mechanosensory and electrosensory receptor organs (neuromasts and ampullary organs, respectively), which originate from common embryonic precursors (lateral line placodes). In the first experiment, Atoh1, a transcription factor that is known to be essential for the development of mechanoreceptors (hair cells) in the ear and lateral line of other vertebrates, was knocked out. As a consequence, neuromasts and ampullary organs in the sterlet lateral lines and the expression of POU4f3 and Gfi1 were lost to varying degrees (this variability is expected due to the mosaicism of gene editing events in F_0_ embryos). This demonstrates that Atoh1 is required for the development of electrosensory as well as mechanosensory receptor cells in the lateral line. It further suggests that POU4f3 and Gfi1, which are known to promote hair cell development in the inner ear downstream of Atoh1, may play a similar role in the development of electroreceptors.

Using the same approach, the study next investigated the effects of knocking out NeuroD4, a transcription factor with electrosensory specific expression, and FoxG1, a transcription factor with mechanosensory specific expression. While NeuroD4 knockout embryos were normal (possibly due to redundant functions of other NeuroD genes shown to be expressed in similar patterns), FoxG1 embryos exhibited losses of neuromasts. In addition, ampullary organs were seen ectopically in areas normally occupied by neuromasts. The authors conclude from this that FoxG1 represses ampullary organ development and suggest that the latter may be the default fate, which needs to be repressed in primordia of mechanoreceptors.

The study is very well conducted, written and documented and provides novel information on the regulation of sensory development in vertebrates. Since none of the established model organisms (e.g. *Xenopus* or zebrafish) possess electroreceptors, it uses the sterlet and successfully establishes CRISPR-based gene editing techniques in this non-model vertebrate organism. Using this technique, it clearly demonstrates a requirement for Atoh1 for the development of both electrosensory and mechanosensory cells and suggests that similar Atoh1 target genes are involved. Taken together with similarities in signal transduction mechanism, these findings strengthen the hypothesis that hair cells and electroreceptors may be evolutionarily closely related to "sister cell types". The study further establishes a new role of FoxG1 in the development of lateral line mechanoreceptors.

The latter finding, however, does not necessarily support the authors' conclusion that FoxG1 represses ampullary organ development and that the latter is the default fate of lateral line receptors. The experiments really only demonstrate that FoxG1 is required for neuromast development in sterlets. The observation of ectopic ampullary organs in these embryos may be explained in several possible ways. There may, for example, be (1) a transformation of fate from neuromasts to ampullary organs due to the loss of FoxG1 as a repressor of ampullary fate (the authors' hypothesis); or (2) an expansion of ampullary precursors because of apoptosis of neuromasts; or (3) an expansion of ampullary precursors because of the absence of neuromasts that exert a repressive effect on ampullary organ development. The patterning disturbances observed in the embryos seem to be more in line with the latter scenario (FoxG1 promotes neuromasts; neuromasts repress ampullary organs), while the fact that FoxG1 seems to be expressed in neuromasts only in electroreceptive fishes (line 567) is more in accordance with the first scenario. Ultimately, this question can only be resolved experimentally (e.g. by overexpressing FoxG1 in electroreceptors; transplanting neuromasts into ampullary fields etc.), which will probably not be feasible in this non-model organism. I therefore, suggest that the authors discuss these various possible interpretations more carefully and tone down their claim of electroreceptors as default fate.

*Reviewer #3 (Recommendations for the authors):*

Minarik and colleagues examine the effects of knockdown of the transcription factor genes Atoh1 and Foxg1 on the development of the mechanosensory and electrosensory lateral line systems in the sterlet (Acipenser ruthenius), a species of sturgeon. Recent work from the same group has demonstrated expression of Atoh1 and its targets Gfi1 and Pou4f3 in both sets of lateral line organs. These genes are highly conserved and necessary for mechanosensory hair cell development. Using CRISPR gene inactivation, the authors demonstrate that disruption of Atoh1 function results in loss of expression of Gfi1 and Pou4f3, as well as differentiation markers of both mechanosensory and electrosensory organs. Expression of the mechanosensory lateral line marker *Sox2* suggests that organs still form but that mechanoreceptors are missing. It is not clear whether ampullary organs still form, but lack electroreceptors, after Atoh1 inactivation.

By contrast targeting the electrosensory-specific transcription factor Neurod4 has no detectable effect, hypothesized due to compensation by other family members. Consistent with this idea, the authors show expression of Neurod1 and Neurod6 in the lateral line; however, they do not examine expression of these genes in the context of Neurod4 inactivation.

Finally, the authors provide evidence for ectopic ampullary organ differentiation after inactivation of the mechanosensory lateral line specific transcription factor Foxg1. These findings could be better documented with multicolor expression analysis, higher power imaging or other approaches to specifically differentiate endogenous organs from ectopic ones. In addition, it is not clear how frequently the ectopic electroreceptors are found. The authors hypothesize that Foxg1 suppresses electroreceptor organ differentiation and speculate that it is therefore a default fate. However, they do not examine expression of mechanoreceptor markers, providing more specific evidence for fate conversion. More evidence is needed to support the idea of a default fate.

The authors suggest that other Neurod family members compensate for disruption of Neurod4. This hypothesis would be further supported by examining the expression of these other genes in Neurod4 crispants.

It is somewhat challenging for the non-expert to distinguish between the different receptors, particularly by location, in the figures provided. While the *Sox2* immunoreactivity should be a helpful landmark, it is difficult to distinguish *Sox2* immunoreactivity with the ISH signal as both appear a similar color. Could alternative methods, including fluorescence, be used to determine whether organs are double-positive? Or could higher power images more clearly distinguish signals or different organ morphologies?

While the loss of organs is clearly documented, the addition of new or ectopic ampullary organs after FoxG1 inactivation is difficult to distinguish. How many new organs are there overall? Are they still *Sox2*^+^?

In addition, the exact phenotype is not clear – has there been loss of neuromasts or neuromast markers (e.g. Isl1, Hmx2 and *Rorb*) – that is, conversion of neuromasts to ampullary organs? Or just the addition of new ectopic ampullae?

Although a very common error, the use of F_0_ to describe CRISPR mutagenesis should be avoided, as in genetic nomenclature F stands for the filial (child) generation. G0 should instead be used.

---

## [Author Response]

Essential Revisions:The reviewers agree that using Cas9/Crispr technology on the sterlet is a significant advancement of this unconventional model system on the highly interesting question of the development and evolution of mechano-/electroreceptors.While the Atoh1 manipulation is overall well documented and convincing, the reviewers think that the following experiments are critical for a more solid manuscript:1) Quantification and statistical assessment of results

As described in more detail below, we have undertaken a more quantitative analysis of both the *Atoh1* and *Foxg1* crispant phenotypes. As part of this analysis, we performed post-ISH *Sox2* immunostaining on all *Foxg1* crispants (previously only done for a subset) to label all neuromasts and prepared skin-mounts. This revealed two additional phenotypes, thus providing greater insight into the potential roles of Foxg1 in lateral line organ development. We have added two new Supplementary Files with the scoring and summary data broken down by individual crispant; three new figures showing the new *Foxg1* data (Figure 6, Figure 7—figure supplement 1, Figure 8) plus new Figure 9 showing summary graphs of the *Foxg1* scoring data (see below for more details). Overall, this was an invaluable exercise that has strengthened the paper.

*(i) Atoh1*: We scored all *Atoh1* crispants for severity of phenotype on each side of the head, defining "severe" as > 2/3 of cranial hair cells/electroreceptors absent in lateral view; "moderate" as between 1/3 and 2/3 absent; and "mild" as <1/3 absent. These data are broken down by individual crispant in new Supplementary File 2. We have added a new Supplementary Figure (Figure 3—figure supplement 2) showing examples of different phenotypes in *Atoh1* crispants, including skin-mount images after post-ISH *Sox2* immunostaining. The outcome is described in the text on lines 394-415 (paragraph starting “Overall, across all riboprobes…”). Briefly, almost two-thirds of *Atoh1* crispants showed a mosaic absence of hair cells/electroreceptors; of these, 80% were classed as "severe", showing that the *Atoh1* sgRNA combination seemed to be relatively efficient and that the phenotype, when present, was generally strong.

*(ii) Foxg1*: To enable a more quantitative approach to analysing the *Foxg1* crispant phenotypes, we undertook post-ISH *Sox2* immunostaining on all *Foxg1* crispants (before, this had only been done for a subset). Excitingly, this revealed additional phenotypes:

(a) In several cases where small patches of *Kcnab3* or *Mafa*-positive electroreceptors were located in the expected position of neuromast lines, post-ISH *Sox2* immunostaining and skin-mount analysis showed that the ectopic electroreceptors had formed within otherwise *Kcnab3* or *Mafa*-negative neuromasts (examples shown in new Figure 8). This suggests that *Foxg1* also acts (directly or indirectly) to repress electroreceptor formation within developing neuromasts.

(b) Gaps were often seen in cranial neuromast lines, without accompanying ectopic ampullary organs (examples shown in new Figure 6, which includes some panels from original Figure 3). This 'missing neuromast' phenotype suggests that mechanosensory-restricted Foxg1 is necessary for the formation and/or maintenance of neuromasts. When we submitted our original manuscript, *Foxg1* expression had not been reported in developing neuromasts in either *Xenopus* or zebrafish (which only have the mechanosensory lateral line). However, since then, a paper was published reporting the expression and function of zebrafish *Foxg1* in the migrating lateral line primordium and during neuromast formation and regeneration (Bell et al., 2024; DOI 10.1242/bio.060580). We have added this citation (lines 138, 813, 892, 895) and removed our previous statement and associated citations about lateral line expression of *Foxg1* being restricted to electroreceptive fishes (original lines 566-70).

We took a more quantitative approach to analysing the *Foxg1* phenotypes by scoring, for each of the four pre-otic neuromast lines (supraorbital, infraorbital, otic, preopercular) on each side of the head in 87 *Foxg1* crispants, the severity of disruption (where "severe" indicates > 2/3 of the line was disrupted; "moderate", between 1/3 and 2/3 of the line was disrupted; "mild", < 1/3 of the line was disrupted) and the type of disruption (ectopic ampullary organs, ectopic electroreceptors within neuromasts; gaps where neuromasts are missing). The new Supplementary File 3 shows the scoring breakdown by each of these lines on the right and left sides of each crispant, together with summary data. The summary scoring data are described in the text (lines 704-810) and presented as graphs in new Figure 9.

The original conclusion of the Results section was:

"Overall, these data suggest that mechanosensory-restricted Foxg1 acts to repress the formation of ampullary organs and electroreceptors within neuromast lines."

This has now been updated as follows:

"Taken together, these data suggest that mechanosensory-restricted Foxg1 is necessary for the formation and/or maintenance of neuromasts (as recently reported in zebrafish; Bell et al., 2024) and also acts (whether directly or indirectly) to repress the formation of ampullary organs and electroreceptors within neuromast lines and electroreceptors within neuromasts."

2) The understanding of the phenotypes could be enhanced with better visual guidance, i.e. adding illustrative cartoons that depict the sensory structures and more "zoom-ins".

As described in more detail below, we have undertaken a more quantitative analysis of both the *Atoh1* and *Foxg1* crispant phenotypes. As part of this analysis, we performed post-ISH *Sox2* immunostaining on all *Foxg1* crispants (previously only done for a subset) to label all neuromasts and prepared skin-mounts. This revealed two additional phenotypes, thus providing greater insight into the potential roles of Foxg1 in lateral line organ development. We have added two new Supplementary Files with the scoring and summary data broken down by individual crispant; three new figures showing the new *Foxg1* data (Figure 6, Figure 7—figure supplement 1, Figure 8) plus new Figure 9 showing summary graphs of the *Foxg1* scoring data (see below for more details). Overall, this was an invaluable exercise that has strengthened the paper.

*(i) Atoh1*: We scored all *Atoh1* crispants for severity of phenotype on each side of the head, defining "severe" as > 2/3 of cranial hair cells/electroreceptors absent in lateral view; "moderate" as between 1/3 and 2/3 absent; and "mild" as <1/3 absent. These data are broken down by individual crispant in new Supplementary File 2. We have added a new Supplementary Figure (Figure 3—figure supplement 2) showing examples of different phenotypes in *Atoh1* crispants, including skin-mount images after post-ISH *Sox2* immunostaining. The outcome is described in the text on lines 394-415 (paragraph starting “Overall, across all riboprobes…”). Briefly, almost two-thirds of *Atoh1* crispants showed a mosaic absence of hair cells/electroreceptors; of these, 80% were classed as "severe", showing that the *Atoh1* sgRNA combination seemed to be relatively efficient and that the phenotype, when present, was generally strong.

*(ii) Foxg1*: To enable a more quantitative approach to analysing the *Foxg1* crispant phenotypes, we undertook post-ISH *Sox2* immunostaining on all *Foxg1* crispants (before, this had only been done for a subset). Excitingly, this revealed additional phenotypes:

(a) In several cases where small patches of *Kcnab3* or *Mafa*-positive electroreceptors were located in the expected position of neuromast lines, post-ISH *Sox2* immunostaining and skin-mount analysis showed that the ectopic electroreceptors had formed within otherwise *Kcnab3* or *Mafa*-negative neuromasts (examples shown in new Figure 8). This suggests that *Foxg1* also acts (directly or indirectly) to repress electroreceptor formation within developing neuromasts.

(b) Gaps were often seen in cranial neuromast lines, without accompanying ectopic ampullary organs (examples shown in new Figure 6, which includes some panels from original Figure 3). This 'missing neuromast' phenotype suggests that mechanosensory-restricted Foxg1 is necessary for the formation and/or maintenance of neuromasts. When we submitted our original manuscript, *Foxg1* expression had not been reported in developing neuromasts in either *Xenopus* or zebrafish (which only have the mechanosensory lateral line). However, since then, a paper was published reporting the expression and function of zebrafish *Foxg1* in the migrating lateral line primordium and during neuromast formation and regeneration (Bell et al., 2024; DOI 10.1242/bio.060580). We have added this citation (lines 138, 813, 892, 895) and removed our previous statement and associated citations about lateral line expression of *Foxg1* being restricted to electroreceptive fishes (original lines 566-70).

We took a more quantitative approach to analysing the *Foxg1* phenotypes by scoring, for each of the four pre-otic neuromast lines (supraorbital, infraorbital, otic, preopercular) on each side of the head in 87 *Foxg1* crispants, the severity of disruption (where "severe" indicates > 2/3 of the line was disrupted; "moderate", between 1/3 and 2/3 of the line was disrupted; "mild", < 1/3 of the line was disrupted) and the type of disruption (ectopic ampullary organs, ectopic electroreceptors within neuromasts; gaps where neuromasts are missing). The new Supplementary File 3 shows the scoring breakdown by each of these lines on the right and left sides of each crispant, together with summary data. The summary scoring data are described in the text (lines 704-810) and presented as graphs in new Figure 9.

The original conclusion of the Results section was:

"Overall, these data suggest that mechanosensory-restricted Foxg1 acts to repress the formation of ampullary organs and electroreceptors within neuromast lines."

This has now been updated as follows:

"Taken together, these data suggest that mechanosensory-restricted Foxg1 is necessary for the formation and/or maintenance of neuromasts (as recently reported in zebrafish; Bell et al., 2024) and also acts (whether directly or indirectly) to repress the formation of ampullary organs and electroreceptors within neuromast lines and electroreceptors within neuromasts."

3) The authors need to consider alternative explanations for their phenotypes more seriously. Specifically, at least three different scenarios could explain the current findings:a) a transformation of fate from neuromasts to ampullary organs due to the loss of FoxG1 as a repressor of ampullary fate (the authors' hypothesis); or (2) an expansion of ampullary precursors because of apoptosis of neuromasts; or (3) an expansion of ampullary precursors because of the absence of neuromasts that exert a repressive effect on ampullary organ development.A better documentation of the Foxg1 effects on patterning, as well as an extended discussion of the different possible scenarios will be helpful. Please see the individual reviews of reviewers 2 and 3 for details on this.

As described in the response to Essential Revision 1, our detailed analysis after post-ISH *Sox2* immunostaining all *Foxg1* crispants revealed gaps within neuromast lines, i.e., loss of neuromasts as an additional phenotype (separate from ectopic ampullary organs). Thus, *Foxg1* plays a role in normal neuromast development and/or maintenance, as reported for zebrafish by Bell et al. (2024) after our original manuscript was submitted. The new analysis, together with skin-mounts, also revealed small patches of *Kcnab3*-positive or *Mafa*-positive electroreceptors within otherwise *Kcnab3*-negative and *Mafa*-negative neuromasts in uninterrupted neuromast lines, suggesting that Foxg1 also represses the formation of electroreceptors within neuromasts.

In the Discussion, we now discuss the various scenarios that could explain the *Foxg1* phenotypes, and cite another recent paper that reported both cell-autonomous and non-cell-autonomous roles for Foxg1 in regulating glial specification in mouse cortex (Bose et al., 2025, DOI: 10.7554/*eLife*.101851):

lines 921-43: "The formation of ectopic ampullary organs/ electroreceptors within neuromast lines in *Foxg1* crispants could be explained by various scenarios. Foxg1 might act as a direct repressor of electrosensory fate, its loss leading to ampullary organ formation instead of neuromasts (and to ectopic electoreceptor formation within neuromasts, as we observed in a few cases). Alternatively, given that ampullary organs normally form later than neuromasts, and neuromasts are missing in the absence of Foxg1 (see also Bell et al., 2024), the expansion of ampullary organ fields could be caused indirectly by the loss of inhibitory signals from neuromasts that normally repress ampullary organ development. The fact that we often saw sections of missing neuromasts without ectopic ampullary organs might argue against this indirect role. However, the "missing neuromasts only" phenotype was primarily seen in the otic neuromast line, where ampullary organ-promoting signals might be more restricted, as the associated ampullary organ field is small and only forms on the dorsal side of part of the line (see schematic in Figure 9A). It is also possible that Foxg1 acts via both of these hypothetical mechanisms. Consistent with this possibility, Foxg1 was recently reported to play a dual role in regulating neurogenesis *versus* gliogenesis in the cortex (Bose et al., 2025). In cortical progenitors, Foxg1 maintains neurogenesis and suppresses gliogenesis cell-autonomously by repressing expression of *Fgfr3*, encoding a receptor for the pro-gliogenic FGF signalling pathway (Bose et al., 2025). In post-mitotic neurons, Foxg1 regulates (directly or indirectly) the expression of FGF ligand genes, suggesting a non-cell-autonomous role for neuron-expressed Foxg1 in regulating gliogenesis in progenitors via FGF signalling (Bose et al., 2025). Thus, it is possible that Foxg1 plays both cell-autonomous and non-cell-autonomous roles in regulating sensory organ and receptor cell formation in the central zone of lateral line sensory ridges."

We have also added some new text immediately before the Summary and Perspective section, to reflect the need for further experiments to test the different scenarios (new text in red font below):

lines 953-7: "To test these hypotheses directly, it will be important in the future to identify global changes in gene expression and chromatin accessibility in the absence of *Foxg1*, as well as analyse sensory cell identity within individual lateral line organs using multiple molecular markers across a range of developmental stages."

Regarding the request for better documentation of the Foxg1 effects on patterning: As noted in the response to Essential Revision 1, we undertook post-ISH *Sox2* immunostaining on all *Foxg1* crispants and scored the severity and type of disruption for each of the four pre-otic neuromast lines (supraorbital, infraorbital, otic, preopercular) on each side of the head in 87 *Foxg1* crispants. The data are provided in new Supplementary File 3, described in the text (lines 704-810) and graphed in new Figure 9. As described in the response to Essential Revision 2, we have also provided better visual documentation of the *Foxg1* effects on patterning, including higher-power views and images of skin-mount preparations, plus a cartoon schematics summarising the various phenotypes.

Reviewer #1 (Recommendations for the authors):In this study, the authors used CRISPR/Cas9-mediated mutagenesis in the sterlet, a small sturgeon species, to investigate the molecular mechanisms underlying the development of electroreceptors. The authors have provided high-quality data, and the paper is well-written with clarity.The authors' key findings are:Atoh1, a well-known 'hair cell' transcription factor, is required for the differentiation of both hair cells and electroreceptors, suggesting deep conservation of molecular mechanisms between these cell types.The mechanosensory-restricted transcription factor Foxg1 appears to repress the formation of electroreceptors within neuromast lines. This leads the authors to propose the intriguing hypothesis that electrosensory organs may represent the 'default' fate within lateral line sensory ridges in electroreceptive vertebrates. It challenges the assumption that mechanosensory hair cells evolutionarily preceded electroreceptors. If further evidence supports this hypothesis, it could reshape our understanding of the evolutionary origins of these sensory cell types.

We have clarified in the revised manuscript that our speculative hypothesis that electroreceptors may be the 'default fate' within lateral line sensory ridges relates solely to developmental mechanisms. We have made this explicit in the revised manuscript by rewording throughout as "…the 'default' developmental fate…" (last sentence of Abstract; last sentence of Introduction [line 140]) and revising the relevant section of the Discussion as follows:

Discussion lines 948-53 (paragraph immediately before "Summary and Perspective"): "Overall, these data lead us to propose a highly speculative hypothesis, namely that electrosensory organs may be the 'default' developmental fate within lateral line sensory ridges in electroreceptive vertebrates, and that Foxg1 represses this fate, whether directly or indirectly, to enable mechanosensory neuromasts and hair cells to form. (This speculation relates solely to developmental mechanisms: electroreceptors most likely evolved in the vertebrate ancestor via the diversification of lateral line hair cells; see Baker, 2019.)"

I have a few suggestions to consider:It would be beneficial to include quantitative data of the observed phenotypes. Identifying subtle differences between images can be challenging.

Please see response to Essential Revision 1.

The current images could be enhanced with visual aids. I recommend adding illustrative cartoons that depict the sensory structures, like Campbell et al. 2024 bioRxiv publication. This would greatly aid in visual comprehension.

Please see response to Essential Revision 2.

Distinguishing between neuromasts and ampullary organs, particularly the electroreceptor formation within neuromast lines, could be difficult. Close-up images or 'zoom-ins' in areas where differentiation is crucial would be extremely helpful.

Please see response to Essential Revision 2.

Reviewer #2 (Recommendations for the authors):This study uses CRISPR/Cas9-based gene editing to knockout genes encoding three transcription factors with potential roles in lateral line development in F_0_ embryos of the sterlet (Acipenser ruthenus). In this bony fish, related to sturgeons, the lateral line system comprises both mechanosensory and electrosensory receptor organs (neuromasts and ampullary organs, respectively), which originate from common embryonic precursors (lateral line placodes). In the first experiment, Atoh1, a transcription factor that is known to be essential for the development of mechanoreceptors (hair cells) in the ear and lateral line of other vertebrates, was knocked out. As a consequence, neuromasts and ampullary organs in the sterlet lateral lines and the expression of POU4f3 and Gfi1 were lost to varying degrees (this variability is expected due to the mosaicism of gene editing events in F_0_ embryos). This demonstrates that Atoh1 is required for the development of electrosensory as well as mechanosensory receptor cells in the lateral line. It further suggests that POU4f3 and Gfi1, which are known to promote hair cell development in the inner ear downstream of Atoh1, may play a similar role in the development of electroreceptors.Using the same approach, the study next investigated the effects of knocking out NeuroD4, a transcription factor with electrosensory specific expression, and FoxG1, a transcription factor with mechanosensory specific expression. While NeuroD4 knockout embryos were normal (possibly due to redundant functions of other NeuroD genes shown to be expressed in similar patterns), FoxG1 embryos exhibited losses of neuromasts. In addition, ampullary organs were seen ectopically in areas normally occupied by neuromasts. The authors conclude from this that FoxG1 represses ampullary organ development and suggest that the latter may be the default fate, which needs to be repressed in primordia of mechanoreceptors.The study is very well conducted, written and documented and provides novel information on the regulation of sensory development in vertebrates. Since none of the established model organisms (e.g. Xenopus or zebrafish) possess electroreceptors, it uses the sterlet and successfully establishes CRISPR-based gene editing techniques in this non-model vertebrate organism. Using this technique, it clearly demonstrates a requirement for Atoh1 for the development of both electrosensory and mechanosensory cells and suggests that similar Atoh1 target genes are involved. Taken together with similarities in signal transduction mechanism, these findings strengthen the hypothesis that hair cells and electroreceptors may be evolutionarily closely related to "sister cell types". The study further establishes a new role of FoxG1 in the development of lateral line mechanoreceptors.The latter finding, however, does not necessarily support the authors' conclusion that FoxG1 represses ampullary organ development and that the latter is the default fate of lateral line receptors. The experiments really only demonstrate that FoxG1 is required for neuromast development in sterlets. The observation of ectopic ampullary organs in these embryos may be explained in several possible ways. There may, for example, be (1) a transformation of fate from neuromasts to ampullary organs due to the loss of FoxG1 as a repressor of ampullary fate (the authors' hypothesis); or (2) an expansion of ampullary precursors because of apoptosis of neuromasts; or (3) an expansion of ampullary precursors because of the absence of neuromasts that exert a repressive effect on ampullary organ development. The patterning disturbances observed in the embryos seem to be more in line with the latter scenario (FoxG1 promotes neuromasts; neuromasts repress ampullary organs), while the fact that FoxG1 seems to be expressed in neuromasts only in electroreceptive fishes (line 567) is more in accordance with the first scenario. Ultimately, this question can only be resolved experimentally (e.g. by overexpressing FoxG1 in electroreceptors; transplanting neuromasts into ampullary fields etc.), which will probably not be feasible in this non-model organism. I therefore, suggest that the authors discuss these various possible interpretations more carefully and tone down their claim of electroreceptors as default fate.

These are excellent points, though we note that our proposal that electroreceptors may be the 'default' fate within lateral line sensory ridges was presented explicitly as a speculation (Abstract) or speculative hypothesis (Discussion), rather than a conclusion or claim (now "a highly speculative hypothesis", line 948). For the revised manuscript, we undertook a more detailed analysis of *Foxg1* phenotypes after performing post-ISH *Sox2* immunostaining on all *Foxg1* crispants to show neuromasts (originally performed only on a subset). Together with skin-mount analysis, this revealed that in some of the crispants analysed with electrosensory-specific markers *Kcnab3* or *Mafa*, small patches of ectopic electroreceptors in the normal position of neuromast lines were actually located within neuromasts in uninterrupted neuromast lines. This suggested that Foxg1 also acts (whether directly or indirectly) to repress electroreceptor formation within neuromasts.

The analysis also revealed that neuromast lines were often disrupted by gaps, without ectopic ampullary organs, consistent with a role for Foxg1 in neuromast development and/or maintenance. After our original manuscript was submitted, *foxg1a* expression and function were reported in the migrating posterior lateral line primordium and neuromasts of zebrafish (Bell et al., 2024; DOI 10.1242/bio.060580). We now cite this paper and have removed our previous statement and associated citations about lateral line expression of *Foxg1* being restricted to electroreceptive fishes (original lines 566-70). Another relevant recent paper identified both cell-autonomous and non-cell autonomous roles for Foxg1 in regulating glial fate in the mouse cortex (Bose et al., 2025; DOI 10.7554/*eLife*.101851).

Taking all of this together, we have added a discussion of the various scenarios that could explain the *Foxg1* phenotypes (lines 921-43). We also emphasise throughout the revised manuscript that Foxg1 may be acting directly or indirectly: line 31; line 813; lines 869-70 (Discussion subtitle); lines 941-3 ("Thus, it is possible that Foxg1 plays both cell-autonomous and non-cell-autonomous roles in regulating sensory organ and receptor cell formation in the central zone of lateral line sensory ridges"); lines 950-51; line 965.

Reviewer #3 (Recommendations for the authors):Minarik and colleagues examine the effects of knockdown of the transcription factor genes Atoh1 and Foxg1 on the development of the mechanosensory and electrosensory lateral line systems in the sterlet (Acipenser ruthenius), a species of sturgeon. Recent work from the same group has demonstrated expression of Atoh1 and its targets Gfi1 and Pou4f3 in both sets of lateral line organs. These genes are highly conserved and necessary for mechanosensory hair cell development. Using CRISPR gene inactivation, the authors demonstrate that disruption of Atoh1 function results in loss of expression of Gfi1 and Pou4f3, as well as differentiation markers of both mechanosensory and electrosensory organs. Expression of the mechanosensory lateral line marker Sox2 suggests that organs still form but that mechanoreceptors are missing. It is not clear whether ampullary organs still form, but lack electroreceptors, after Atoh1 inactivation.

The supporting-cell marker *Sox2* is expressed in ampullary organs, but at much lower levels than in neuromasts (Figure 1A,B). Even before in situ hybridisation (ISH), ampullary organ labelling is variable and it is weaker still after ISH. However, we detected ampullary organs after post-*Kcnab3* ISH *Sox2* immunostaining in a few *Atoh1* crispants and showed one example in the submitted manuscript:

Original p. 7, lines 244-6: "Post-*Kcnab3* ISH immunostaining for *Sox2* confirmed that ampullary organs were still present, as well as neuromasts (Figure 1O^1^,P^1^)."

In the revised manuscript, we have provided an additional example and re-worded the text to emphasise the weakness and variability of the post-ISH *Sox2* immunostaining in ampullary organs:

lines 321-5: "Post-*Kcnab3* immunostaining for *Sox2*, although very weak and variable in ampullary organs (versus reliable labelling of neuromasts), confirmed in a few cases that ampullary organs were still present, as well as neuromasts (Figure 3O^1^,P^1^; Figure 3—figure supplement 2G-H^1^). Hence, the phenotype in ampullary organs was also specific to receptor cells."

By contrast targeting the electrosensory-specific transcription factor Neurod4 has no detectable effect, hypothesized due to compensation by other family members. Consistent with this idea, the authors show expression of Neurod1 and Neurod6 in the lateral line; however, they do not examine expression of these genes in the context of Neurod4 inactivation.

ISH is non-quantitative and the wild-type expression patterns are not uniform within individual larvae (see the *Neurod* family gene expression figure, now Figure 1—figure supplement 1). Hence, we do not think it would be straightforward to identify changes in the expression levels of *Neurod1* and/or *Neurod6* in ampullary organs in *Neurod4* crispants. (As they are already co-expressed in ampullary organs, we would not expect to see any change in their spatial expression patterns.) Furthermore, if the Neurod family transcription factors act redundantly and are only found in low abundance, then changes in mRNA expression levels might not be needed to compensate for the loss of *Neurod4*. Regardless, we are unable to perform additional experiments at this time.

Finally, the authors provide evidence for ectopic ampullary organ differentiation after inactivation of the mechanosensory lateral line specific transcription factor Foxg1. These findings could be better documented with multicolor expression analysis, higher power imaging or other approaches to specifically differentiate endogenous organs from ectopic ones.

Although multiplex fluorescent HCR would in theory be an ideal approach, in practice the analysis would be challenging owing to the combination of CRISPR mosaicism and the very large size of sterlet larvae. We have in the past attempted double chromogenic ISH with Fast Red as the other colour, but this did not work well. In any case, we are unable to perform additional experiments at this time. However, in order to document the findings better, we have provided more higher-power views and prepared and imaged skin mounts of post-ISH *Sox2* immunostained *Foxg1* crispants (see new Figure 6, new Figure 7—figure supplement 1; new Figure 8).

In addition, it is not clear how frequently the ectopic electroreceptors are found.

In our new detailed analysis of *Foxg1* crispants after post-ISH *Sox2* immunostaining, we scored the level of severity (proportion of the neuromast line disrupted) and type of disruption (ectopic ampullary organs, electroreceptors within neuromasts, neuromast gaps) for each of the pre-otic neuromast lines (supraorbital, infraorbital, otic, preopercular) on each side of the head in 87 *Foxg1* crispants. As described in the revised text:

"Overall, 64% (n=56/87) of *Foxg1* crispants showed a phenotype (Supplementary File 3). In the phenotypic *Foxg1* crispants, neuromast lines were interrupted by ectopic ampullary organs in 75% (n=42/56), by gaps in 86% (n=48/56) and by both in 61% (n=34/56) (Supplementary File 3). Electroreceptors were identified within neuromasts in 13% of phenotypic *Foxg1* crispants (n=7/56; Supplementary File 3)."

We go on to provide a detailed breakdown of the phenotypes for each pre-otic neuromast line across all phenotypic *Foxg1* crispants (starting immediately after text quoted above, "In an attempt to quantify the observed phenotypes further…" and continuing for two further paragraphs; lines 715-801). The scoring data for each line in each crispant, together with summary scoring data, are provided in new Supplementary File 3. Graphs of the summary scoring data are provided in new Figure 9.

The authors hypothesize that Foxg1 suppresses electroreceptor organ differentiation and speculate that it is therefore a default fate. However, they do not examine expression of mechanoreceptor markers, providing more specific evidence for fate conversion. More evidence is needed to support the idea of a default fate.

It would indeed be interesting to examine expression of mechanoreceptor-specific markers, but we are unable to do further experiments at this time. However, as described above, our new detailed analysis after post-ISH *Sox2* immunostaining all *Foxg1* crispants revealed loss of neuromasts as an additional phenotype (separate from ectopic ampullary organs). Thus, *Foxg1* plays a role in normal neuromast development, as reported for zebrafish after our original manuscript had been submitted (Bell et al., 2024, DOI 10.1242/bio.060580). We also identified small patches of electroreceptors within neuromasts in uninterrupted neuromast lines, suggesting Foxg1 also acts to repress electroreceptor formation within neuromasts. We discuss the various scenarios for how Foxg1 might be acting (revised Discussion section "Foxg1 represses electroreceptor formation in the neuromast-forming central zone of lateral line sensory ridges, whether directly or indirectly", 5th paragraph starting “The formation of ectopic ampullary organs/electroreceptors…”). We have also added text reflecting the need for analysis at different stages and with multiple molecular markers in order to test these hypotheses:

Revised Discussion, paragraph immediately preceding "Summary and Perspective", final sentence (lines 953-7: "To test these hypotheses directly, it will be important in the future to identify global changes in gene expression and chromatin accessibility in the absence of *Foxg1*, as well as analyse sensory cell identity within individual lateral line organs using multiple molecular markers across a range of developmental stages.").

The authors suggest that other Neurod family members compensate for disruption of Neurod4. This hypothesis would be further supported by examining the expression of these other genes in Neurod4 crispants.

Please see response to reviewer 2's similar comment above.

We also apologise for a confusing mistake in the original text: as shown in the original Supplementary Figure S5 (now Figure 1—figure supplement 2), *Neurod1* and *Neurod6* are also expressed in ampullary organs (as well as neuromasts), but *Neurod2* was not expressed in any lateral line organs. This has now been corrected in the main text:

lines 475-8: "*Neurod1* and *Neurod6* proved to be expressed in sterlet ampullary organs, as well as neuromasts, while *Neurod2* was not expressed in either (Figure 1—figure supplement 2)."

lines 861-3: *"*However, we found that *Neurod1* and *Neurod6* (but not *Neurod2*) are also expressed in sterlet ampullary organs (as well as neuromasts), suggesting that Neurod4 may act redundantly with one or both of these factors in developing ampullary organs."

It is somewhat challenging for the non-expert to distinguish between the different receptors, particularly by location, in the figures provided. While the Sox2 immunoreactivity should be a helpful landmark, it is difficult to distinguish Sox2 immunoreactivity with the ISH signal as both appear a similar color. Could alternative methods, including fluorescence, be used to determine whether organs are double-positive? Or could higher power images more clearly distinguish signals or different organ morphologies?

Please see response to reviewer 2's similar comment above. Higher power views of post-ISH *Sox2*-immunostained *Foxg1* crispants are now included in the new Figure 6, along with the same regions shown before immunostaining, so the nature of the signal (ISH vs. immunostaining) can be directly checked for individual organs. Also, in the new skin-mount images, the distinctive colours of the two techniques used become clear, with *Sox2* signal being black, versus the blue/purple ISH signal (new Figure 3—figure supplement 2; new Figure 7—figure supplement 1; new Figure 8).

While the loss of organs is clearly documented, the addition of new or ectopic ampullary organs after FoxG1 inactivation is difficult to distinguish. How many new organs are there overall? Are they still Sox2^+^?

As described in the responses to the Essential Revisions and reviewer 2's similar comment, we have tried to make this clearer by providing more higher-power views and preparing and imaging skin mounts of post-ISH *Sox2* immunostained Foxg1 crispants (see new Figure 6, new Figure 7—figure supplement 1; new Figure 8).

Regarding how many new organs there are overall: the mosaicism of Foxg1 phenotypes and variability across crispants means that the absolute number of new/ectopic ampullary organs in each crispant is unlikely to be very informative. However, as described in earlier responses, our more quantitative approach after performing post-ISH *Sox2* immunostaining on all Foxg1 crispants, both identified additional phenotypes and enabled us to score the severity and type of disruption to individual pre-otic neuromast lines on each side of the head (individual crispant data and summary data in new Supplementary File 3; described in the text lines 711-801; graphs in new Figure 9).

Regarding expression of the supporting-cell marker *Sox2* in ectopic ampullary organs: as noted in earlier responses, *Sox2* immunostaining is much weaker and more variable in ampullary organs than neuromasts even before ISH, and the signal is even weaker and more variable after ISH. However, our new skin-mount analyses enabled detection of *Sox2* signal in ectopic ampullary organs in a few Foxg1 crispants, as shown in new Figure 7—figure supplement 1 (panels G and J).

In addition, the exact phenotype is not clear – has there been loss of neuromasts or neuromast markers (e.g. Isl1, Hmx2 and Rorb) – that is, conversion of neuromasts to ampullary organs? Or just the addition of new ectopic ampullae?

As noted in the response to reviewer 2's similar comment above, it is not possible to answer these important questions without analysing Foxg1 crispants at earlier developmental stages and using a range of molecular markers, including mechanosensory-specific markers such as Isl1, Hmx2 and *Rorb* (previously published in Minařík et al., 2024), as proposed here. Unfortunately, we are unable to do further experiments at this time. Furthermore, as described above, our new detailed analysis after post-ISH *Sox2* immunostaining all Foxg1 crispants revealed loss of neuromasts as an additional phenotype (separate from ectopic ampullary organs), as well as small patches of electroreceptors within neuromasts in uninterrupted neuromast lines. As mentioned earlier, the various scenarios for how Foxg1 might be acting are now discussed (lines 921-43) and we have also added text reflecting the need for analysis at different stages and using multiple molecular markers to test these hypotheses:

Resubmission lines 953-7: "To test these hypotheses directly, it will be important in the future to identify global changes in gene expression and chromatin accessibility in the absence of Foxg1, as well as analyse sensory cell identity within individual lateral line organs using multiple molecular markers across a range of developmental stages."

Although a very common error, the use of F_0_ to describe CRISPR mutagenesis should be avoided, as in genetic nomenclature F stands for the filial (child) generation. G0 should instead be used.

Thanks for pointing this out: now corrected throughout.